## RESEARCH ARTICLE

# The mouse neonatal small intestine is regionally specialized for protein absorption and transepithelial transport

Carina L. Block[1,*], Laura Childers[1], Abby L. Cortez[1], Kristina Sakers[1,2], Tylor R. Lewis[3], Daniel S. Levic[1], Lauren C. Frazer[4], Corey M. Jania[4], Vadim Y. Arshavsky[3], Misty Good[4], Cagla Eroglu[1,2,5,*] and Michel Bagnat[1,*]

## ABSTRACT

In neonates, gastric protein digestion is limited, requiring specialized mechanisms for intestinal protein absorption. While neonatal enterocytes are thought to mediate endocytosis, degradation and transcytosis of dietary proteins, whether these activities are spatially segregated and their molecular basis are unknown. Here, we combine *in vivo* and *ex vivo* cargo transport assays with transcriptomic and genetic approaches in mice to uncover distinct roles for jejunal and ileal neonatal enterocytes. We show that the jejunum is highly active in transepithelial transport of intact proteins, whereas the ileum specializes in their lysosomal degradation. Although both regions express similar endocytic receptors, structural and transcriptional analyses uncover divergent endolysosomal programs. Single-cell RNA sequencing reveals that jejunal and ileal enterocytes emerge from a similar progenitor pool but diverge transcriptionally. Moreover, ileal enterocytes share features with lysosome-rich enterocytes in zebrafish, suggesting evolutionary conservation. Conditional loss of Dab2 disrupts protein, but not antibody, transcytosis, supporting distinct uptake routes for nutritional and immune cargos. These findings show regional and functional specialization of enterocytes during early postnatal development, and underscore conserved protein absorption mechanisms in vertebrates.

KEY WORDS: Neonatal enterocyte, Transcytosis, Endocytosis, Nutrition, Gastrointestinal biology, Protein absorption, Mouse

## INTRODUCTION

The small intestine is regionally specialized along the proximal-to-distal axis to support distinct nutrient absorption and physiological functions. Traditionally, this specialization has been loosely divided into three regions from the proximal to distal: the duodenum, jejunum and ileum. Recent advances in spatial transcriptomics have revealed further metabolic distinctions within these regions; however, to date, most studies have been performed in adult tissue (Zwick et al., 2024).

In neonatal mammals, proteins derived from maternal milk provide not only nutritional value but also serve immunological and regulatory roles (Kulkarni and Newberry, 2019; Weström et al., 2020). To support these dual functions, the neonatal gastrointestinal tract is uniquely adapted to preserve bioactive milk proteins while simultaneously meeting the high nutritional demands of rapid growth. Because the neonatal stomach is not yet acidified and has reduced pepsin activity necessary for the proteolysis of proteins, milk proteins arrive at the intestinal lumen largely intact (Henning, 1985).

Both the neonatal jejunum and ileum are capable of endocytosing luminal proteins, which can then either undergo transcellular transport across the epithelium or intracellular degradation, possibly within large lysosomal vacuoles characteristic of distal neonatal enterocytes (Abrahamson et al., 1979; Baba et al., 2002; Gonnella et al., 1987). Because we have previously demonstrated that these vacuolated enterocytes indeed digest luminal proteins intracellularly in zebrafish, we refer to them as lysosome-rich enterocytes (LREs) (Park et al., 2019). Electron microscopy (EM) studies in mammals have revealed some structural similarities between enterocytes in the neonatal jejunum and ileum, including a dense apical endocytic membrane system and abundant coated vesicles (Fujita et al., 2007; Wilson et al., 1991). However, whether these cell populations exhibit functional differences remains unclear.

Previous EM-based, protein-tracing studies found that protein is taken up by both the jejunum and ileum and distributed throughout the endolysosomal network in both regions (Fujita et al., 2007; Gonnella et al., 1987; He et al., 2008; Rodewald and Abrahamson, 1982), with the jejunum also showing tracers present in intercellular spaces. These findings led to the widely held, yet unproven, hypothesis that the neonatal jejunum is specialized for the transcytosis of intact proteins, while the neonatal ileum is primarily involved in protein absorption. However, the lack of quantitative assays and methods for dissecting the respective contributions of each intestinal region, as well as the genetic and cellular basis underlying their physiology remain a limitation, precluding firm conclusions.

Proteins in the intestinal lumen can also be internalized via receptor-mediated endocytosis (RME) – which includes both selective and non-selective pathways – and by fluid-phase endocytosis. RME has been most extensively characterized in the jejunum (Kraehenbuhl and Campiche, 1969). The jejunum expresses high levels of the neonatal Fc receptor (FcRn), which binds immunoglobulin G (IgG) and facilitates its transcytosis across the intestinal epithelium (He et al., 2008; Rodewald, 1973, 1980; Rodewald and Kraehenbuhl, 1984). Other cargos that lack selective receptors, such as HRP, have also been detected within jejunal enterocytes, suggesting utilization of both selective and non-selective forms of RME (Baba et al., 2002).

[1]Department of Cell Biology, Duke University Medical Center, Durham, NC 27710, USA. [2]Howard Hughes Medical Institute, Duke University Medical Center, Durham, NC 27710, USA. [3]Department of Ophthalmology, Duke University School of Medicine, Durham, NC 27710, USA. [4]Division of Neonatal-Perinatal Medicine, Department of Pediatrics, University of North Carolina at Chapel Hill, Chapel Hill, NC 27599, USA. [5]Department of Neurobiology, Duke University Medical Center, Durham, NC 27710, USA.

*Authors for correspondence (carina.block@duke.edu; cagla.eroglu@duke.edu; michel.bagnat@duke.edu)

C.L.B., 0000-0002-2424-2878; C.E., 0000-0002-7204-0218; M.B., 0000-0002-3829-0168

In previous work, we have demonstrated that a conserved endocytic receptor complex comprising Cubilin (Cubn), Amnionless (Amn) and the adaptor protein Disabled-2 (Dab2) is essential for non-selective, receptor-mediated protein uptake (Childers et al., 2025; Park et al., 2019). Cubn is a broad-spectrum scavenger receptor capable of binding diverse protein ligands, including dietary proteins (Amsellem et al., 2010; Nielsen et al., 2016; Park et al., 2019), while Amn is a transmembrane partner that links Cubn to Dab2, which in turn mediates the internalization of bound cargo (Fyfe et al., 2004; He et al., 2005; Maurer and Cooper, 2005; Strope et al., 2004). In zebrafish, the Cubn/Amn/Dab2 complex is expressed in a distal region of the mid-intestine (Park et al., 2019), which is analogous in cellular morphology and function to the mammalian ileum (Lickwar et al., 2017). Genetic elimination of any component of this complex in zebrafish and Dab2 in mice markedly reduces dietary protein uptake by LREs, resulting in stunted growth and malnutrition (Park et al., 2019). Moreover, disruption of this complex impaired trans-epithelial protein transport in zebrafish, supporting the model whereby this machinery serves as a high-capacity pathway for protein transcytosis (Park et al., 2019).

Despite these insights, it remains unclear to what extent this endocytic machinery is expressed and functional in the mammalian intestine. Furthermore, how enterocytes of the neonatal jejunum and ileum differ in molecular function and protein handling remains unknown. In this study, we have investigated how the neonatal small intestine is regionally specialized to support the uptake, degradation and transcytosis of luminal proteins. By integrating *in vivo* and *ex vivo* functional assays with transcriptomic and cell biological analyses, we reveal distinct regional programs for protein handling in the jejunum and ileum. Our findings directly demonstrate that the jejunum plays a dominant role in trans-epithelial protein and soluble cargo transport, while the ileum has a high capacity for intracellular protein degradation. Furthermore, we define the molecular machinery underlying these processes and show that regional specialization is conserved across species and developmentally regulated.

## RESULTS
### Neonatal vacuolated enterocytes can be found in mice and humans
Neonatal enterocytes with the capacity for protein uptake have been shown to have distinctive vacuoles (Fujita et al., 2007; Park et al., 2019). To map localization of enterocytes with vacuoles along the mouse GI tract, we performed periodic acid–Schiff (PAS) staining on postnatal day 6 (P6) and day 36 (P36) samples from the duodenum, jejunum and ileum. Lysosomal vacuoles are enriched in glycoproteins, which stain magenta with PAS. As expected, no vacuoles were detected in the post-weaning intestine at P36 (Fig. 1A,B). By contrast, in pre-weaning neonatal mice at P6, vacuoles were detected in both the jejunum and ileum, with higher abundance in the ileum (Fig. 1A,B). Notably, these cells differed in their PAS staining intensity (Fig. 1A, black arrowheads). Enterocytes with vacuoles are readily distinguishable from goblet cells, which exhibit uniform deep-magenta staining due to their mucin content (Fig. 1A, magenta arrowheads).

In humans, LREs have only been reported prenatally (Weström et al., 2002). Given the spatial and developmental distinctions we observed in neonatal mice, we reasoned that vacuolated enterocytes (LREs) may have been overlooked in human infants due to limited tissue availability and the substantial length of the human intestine at birth (~3 m) (Weaver et al., 1991). To explore this possibility, we examined ileal tissue from human infants who had undergone surgical resection for necrotizing enterocolitis (NEC). PAS staining

revealed the presence of vacuoles in neonatal human ileum enterocytes (Fig. 1C, arrowheads). The sample shown is from an infant with a postmenstrual age of 58 weeks (approximately 30 weeks after birth), at which point enterocytes exhibited intermediate morphology between neonatal and mature enterocytes (Fig. 1A). While PAS staining allowed us to distinguish enterocytes with vacuoles from goblet cells (Fig. S1A), we further confirmed their identity by performing EM on a separate sample from a neonate at 35 weeks postmenstrual age (12 weeks after birth). EM distinguishes these two cell types morphologically: goblet cells contain densely packed mucin granules, whereas vacuoles can be identified by a spherical shape with a single membrane and by varying electron density. This distinction was evident in the neonatal human sample, where we identified an enterocyte with a prominent lysosome, characteristic of LREs (Fig. 1D), which was clearly distinct from neighboring goblet cells (Fig. S1B). These data show that LREs are present in the human neonatal ileum and suggest they can persist up to at least 4 months of age.

### The neonatal ileum is specialized for rapid protein uptake and degradation
Previous studies have reported protein uptake in both the neonatal jejunum and ileum (Baba et al., 2002; Bara et al., 2022). To examine regional differences in this process, we gavaged neonatal mice with mCherry, a pH-insensitive fluorescent protein that can be readily visualized by fluorescence microscopy. This gavage technique deposits cargo directly into the stomach, allowing it to transit naturally through the small intestine. To ensure sufficient passage from proximal to distal regions, we harvested the intestines 3 h post-gavage and analyzed mCherry uptake using confocal microscopy on 14 equal proximal-to-distal segments (Fig. 1E; Fig. S2). While minimal mCherry uptake was observed in the duodenum (Fig. 1F, left), high levels of uptake were detected in the jejunum and ileum (Fig. 1F, middle and right panels). To control for autofluorescence, we also analyzed mice gavaged with PBS, which produced no detectable signal (Fig. S3A, left). Additionally, post-weaning mice gavaged with mCherry also showed no signal, indicating that protein uptake is specific to the neonatal period (Fig. S3A, right).

Although previous studies have reported protein uptake in the neonatal jejunum (Baba et al., 2002; Bara et al., 2022; Ono, 1975; Weström et al., 2002), the high level of mCherry accumulation in this region is especially notable in comparison to the ileum. The increased mCherry signal in the jejunum compared to the ileum prompted us to consider multiple possibilities, including regional differences in endocytic uptake, protein degradation or recycling dynamics, as well as potential confounding variables such as intestinal transit time and local protein concentration *in vivo*. To isolate protein uptake capacity from confounding *in vivo* variables, we developed an *ex vivo* closed-loop assay. Segments of neonatal jejunum and ileum were isolated, filled with fluorescent cargo, ligated at both ends and incubated in intestinal media for 30 min (Fig. 1G). In this assay, both regions internalized mCherry protein (Fig. 1G), which can be detected with minimal autofluorescence (Fig. S3B,C). In the jejunum, mCherry was largely restricted subapically, while in the ileum, it localized to both apical endosomes and large lysosomal vacuoles (Fig. 1G, magenta arrowheads), consistent with the faster kinetics of internalization and lysosomal transport (Fig. 1H). These results suggest that the ileum has a greater capacity for protein uptake than the jejunum.

In addition to an increased uptake rate, the ileum may also have enhanced protein degradation capacity. To directly test this, we returned to the *in vivo* gavage system and co-gavaged mCherry and

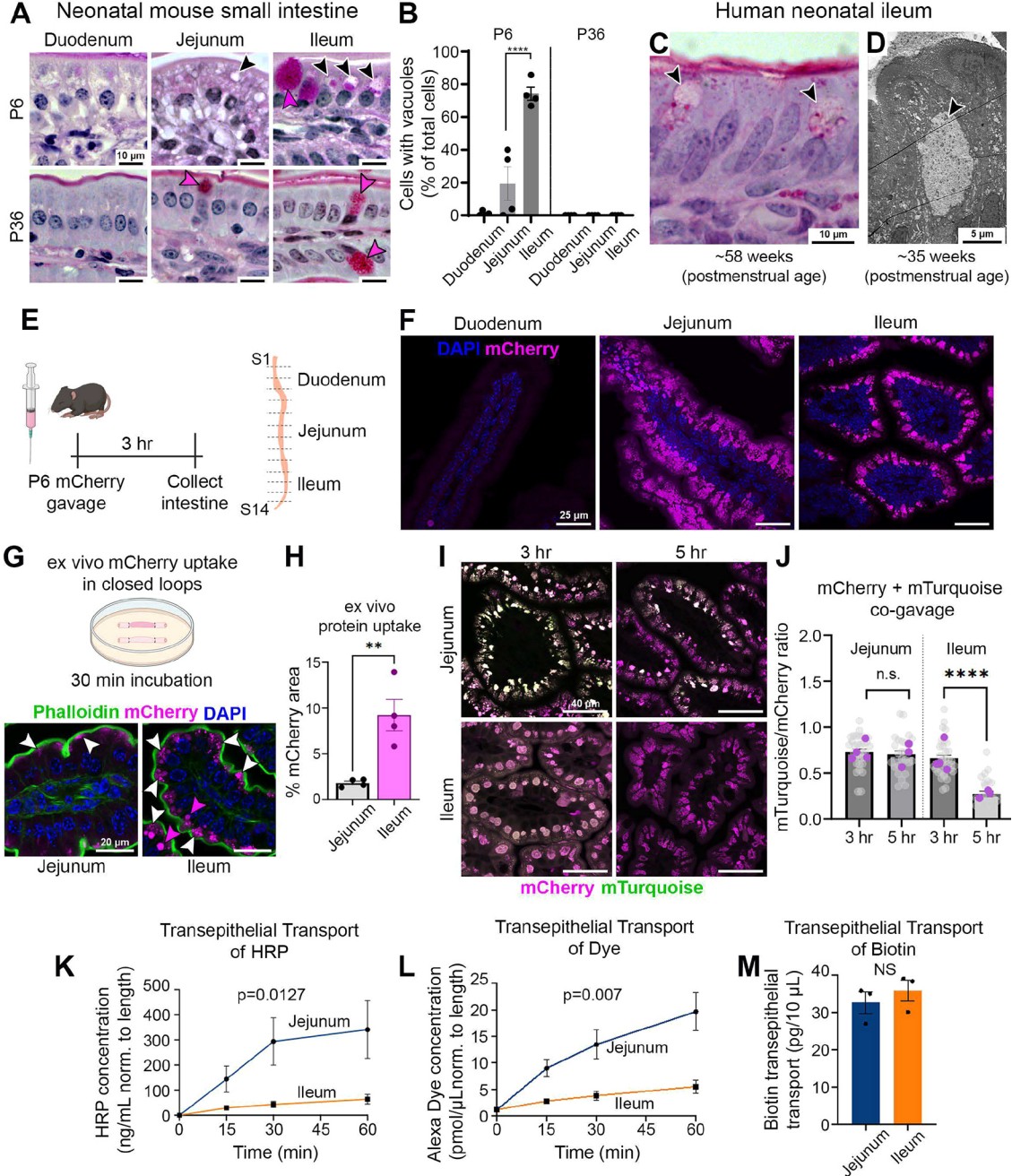

**Fig. 1. Neonatal enterocytes differ in their uptake kinetics and are regionally specialized for transcytosis or degradation.** (A) PAS staining in the small intestine of P6 and P36 mice showing vacuolated enterocytes (black arrowheads) and goblet cells (magenta arrowheads). (B) Quantification of vacuolated enterocytes ($n$=4 mice/age, one-way ANOVA with Bonferroni post-hoc tests; ****$P$<0.0001). (C) Vacuolated cells in the ileum of a human infant (black arrowheads) at 58 weeks. (D) EM of a vacuolated enterocyte (arrowhead) from a neonatal human at 35 weeks. (E) Gavage approach in the mouse small intestine. (F) Images of intestinal segments showing mCherry signal 3 h post-gavage. Displayed images are also found in Fig. S2, detailing more granular regional differences in protein uptake. (G) Ex vivo closed-loop assay and images of mCherry uptake in P3 intestinal samples 30 min after incubation. White arrowheads indicate subapical accumulation; magenta arrowheads indicate vacuole accumulation. (H) Quantification of mCherry signal ($n$=4 animals/group, unpaired $t$-test; **$P$=0.0052). (I) Images from co-gavage of mCherry and mTurquoise at 3 and 5 h post-gavage in P6 mice. (J) Quantification of the ratio of mTurquoise to mCherry ($n$=4 animals/time-point, unpaired $t$-test; ****$P$<0.0001). (K-M) Quantification of transcytosis of HRP, Alexa dye and biotin in ex vivo closed loop assays. Biotin was measured at 15 min [$n$=3 animals/group, repeated measures two-way ANOVA (L,M) and unpaired $t$-test (N)]. $P$-values correspond to main effect of region. Data are mean±s.e.m. Panels E and G created in BioRender by Eroglu, C., 2025. https://BioRender.com/xly5yrw. This figure was sublicensed under CC BY 4.0 terms.

mTurquoise – two fluorescent proteins that are pH insensitive but have significantly different degradation kinetics (half-lives of 2.7 and 0.23 h, respectively) (Park et al., 2019). We then compared the mTurquoise/mCherry fluorescence ratio at 3 and 5 h post-gavage (Fig. 1I). In compartments with high proteolytic activity, such as a lysosome, this ratio is expected to decline over time due to the faster

degradation of mTurquoise compared to mCherry. At 3 h post-gavage, both fluorescent proteins were robustly detected in the jejunum and ileum, with no significant regional differences. By 5 h, the mTurquoise/mCherry ratio remained stable in the jejunum, indicating it was not within a digestive compartment. On the other hand, the ratio decreased by ~50% in the ileum, indicating

enhanced lysosomal degradation (Fig. 1I,J). Together, these findings demonstrate that the neonatal ileum has a significantly higher capacity for both protein uptake and degradation than the jejunum. These results further suggest that the proteins internalized in the jejunum are at least in part processed through non-degradative intracellular pathways.

## The neonatal jejunum mediates trans-epithelial transport of luminal cargos

Previous studies have established that the neonatal jejunum is able to internalize maternal antibodies via the neonatal Fc receptor (FcRn) and transport them across the epithelial barrier (He et al., 2008). However, it remains unclear whether proteins internalized by alternative pathways – such as non-specific RME or fluid-phase endocytosis – are also directed toward transcytosis. To address this question and investigate regional differences, we used our *ex vivo* closed-loop approach to assay cargos internalized via non-FcRn pathways. To do so, we filled jejunal and ileal segments with HRP, a 40 kDa protein that is internalized via non-selective RME (Baba et al., 2002). We first injected intestinal loops with 50 µg of HRP and then ligated, washed and incubated them in intestinal media. We then collected media at 0, 15, 30 and 60 min to assess transcytotic activity over time. Remarkably, jejunal segments transcytosed approximately five times more HRP than those of the ileum (Fig. 1K). These findings suggest that, in addition to its well-established role in antibody transport, the neonatal jejunum is also capable of transcytosing other proteins at significantly higher levels than the ileum. This functional difference also extended to non-protein cargo: jejunal segments also demonstrated greater transport of soluble dye, which is internalized via fluid-phase endocytosis (Fig. 1L). Importantly, though the ileum transported far less protein and dye compared to the jejunum, there was a measurable amount of HRP and dye transported across the intestinal epithelium (Fig. 1K,L). In contrast, biotin – a water-soluble vitamin transported via sodium-dependent multivitamin transporters (SMVTs) – showed no significant difference in transport between the neonatal jejunum and ileum, reaching maximum saturation after 15 min of incubation (Fig. 1M).

Collectively, these findings indicate that the neonatal jejunum is not only responsible for FcRn-mediated antibody transcytosis but is also uniquely equipped to transcytose a broad range of protein and non-protein soluble cargos. The regional specificity observed in our functional assays likely contributes to the differential accumulation of macromolecules such as mCherry in the neonatal jejunum in our *in vivo* assay.

## Transcriptomic profiling reveals region- and age-specific programs

The results of our transport and degradation assays raised the question of whether non-selective RME of luminal proteins depends on shared or distinct endocytic machinery in the jejunum versus the ileum. To further investigate the molecular basis of functional differences we observed, we turned to transcriptomic profiling. While several studies have explored regional differences in gene expression in the adult small intestine, comparable analysis in neonates remains limited (Zwick et al., 2024). To fill this gap, we performed bulk RNA sequencing on isolated epithelial cells from the jejunum and ileum at neonatal and postweaning time points (P6 and P36) (Fig. S4A). Principal component analysis (PCA) showed that samples clustered by both age and region (Fig. 2A). Differential gene expression analysis revealed that regional differences were most pronounced at P6, with more than twice as many genes differentially expressed between the jejunum and ileum at this age

compared to P36 (1099 versus 488 genes, respectively; Table S1, Fig. S4B). As expected, we also observed substantial developmental changes, with 2348 genes differentially expressed between P6 and P36 in the jejunum and 2561 in the ileum (Fig. S4B,C).

To define gene expression programs associated with developmental stage, we identified genes that were consistently up- or downregulated with age in both regions (Fig. S4D). KEGG pathway analysis of these shared gene sets revealed that genes upregulated at P36 were enriched in pathways related to arachidonic and linoleic acid metabolism (Fig. 2B). Additionally, we observed increased expression of genes in the intestinal immune network for IgA production, consistent with postnatal immune maturation. In contrast, genes upregulated at P6 – when LREs are abundant – were enriched for lysosomal and degradative pathways, suggesting enhanced lysosomal function in both the neonatal jejunum and ileum compared to the adult intestine (Fig. 2B).

To investigate regional differences in the neonatal intestine, we performed differential gene expression analysis between the P6 jejunum and ileum. This analysis identified 946 differentially expressed genes, with 311 upregulated in the jejunum and 635 in the ileum (Fig. S4C). Among the top genes enriched in the ileum were *Afp*, *Cldn8*, *Rarres1*, *Anxa8* and *Cubn*, many of which were specifically enriched at P6, suggesting a developmental role in ileum-specific function (Fig. 2C). In the jejunum, top upregulated genes compared to the P6 ileum included *Fcgrt*, which mediates maternal antibody transfer, and *Cd36*, which is involved in uptake of long-chain fatty acids (Cifarelli and Abumrad, 2018).

To further dissect these regional differences, we performed KEGG pathway analysis on genes upregulated in each region. In the P6 ileum, we observed enrichment in pathways related to ECM-receptor interactions and complement activation (Fig. 2D). Pathways related to protein digestion and absorption were also enriched, in agreement with both the literature and our findings in Fig. 1. In contrast, the P6 jejunum showed enrichment in cholesterol metabolism, PPAR signaling, lipid processing and absorption, and bile secretion pathways (Fig. 2D, right), consistent with its known role in lipid absorption in neonates (Booth et al., 1961; Weström et al., 2002).

As shown in Fig. 1, both the neonatal jejunum and ileum actively endocytose luminal proteins. We next asked whether this uptake involved the same endocytic machinery previously identified in ileal LREs (Park et al., 2019), including *Dab2*, *Cubn* and *Amn*. Notably, expression of these genes was elevated in both the P6 jejunum and ileum compared to P36 (Fig. S4E). Although gene expression levels were higher in the ileum, this difference was not statistically significant; however, we and others have previously found increased levels of Dab2 protein in the ileum (Park et al., 2019; Vázquez-Carretero et al., 2011). Furthermore, the ileum is enriched for multi-ligand endocytic receptors and adaptors with vesicle trafficking proteins, consistent with high endocytic capacity (Fig. S4F). By contrast, the jejunum shows selective enrichment for transport machinery mediating transcytosis of immunoglobulins and antigens, and for apical-basolateral trafficking regulators (*Rab8b*, *Rab30*, *Rilp*, *Yif1a*, *Ap1s3*, *Clint1*, *Arf4* and *Exoc3l4*; Fig. S4F). These findings suggest that regional specialization in protein handling is likely driven by downstream regulatory mechanisms rather than gene expression levels alone. Supporting this, genes involved in endocytic recycling were specifically upregulated in the P6 jejunum. This includes both *Rab8a* and *Rab8b*, which are crucial for the apical localization of proteins in neonatal enterocytes (Sato et al., 2014, 2007). Conversely, proteases associated with protein degradation were more highly expressed in the ileum (Fig. 2E).

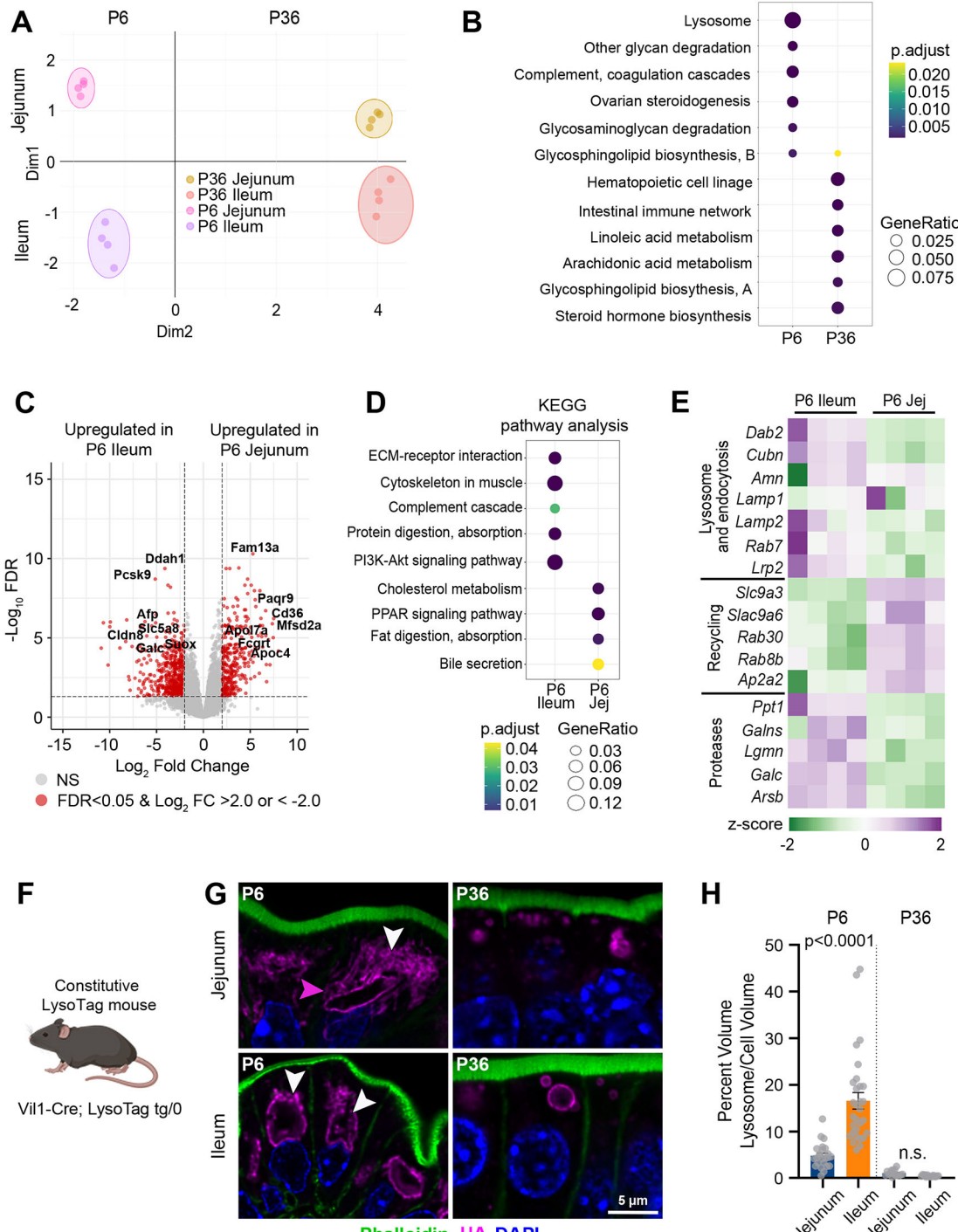

**Fig. 2. Neonatal enterocytes share high lysosomal activity but differ in the endolysosomal system.** (A) PCA plot of intestinal epithelial cells from jejunum and ileum at P6 and P36 (*n*=4 animals/age). (B) KEGG pathway analysis of genes that are upregulated in both the jejunum and ileum at P6 and P36. (C) Volcano plot shows differentially expressed genes (in red) between P6 jejunum and ileum. FDR<0.05 and a log fold change of <−2.0 or >2.0. (D) KEGG pathway analysis of genes that are differentially expressed between P6 jejunum and P6 ileum. (E) Heatmap of DEGs. (F) Lysotag mice were generated by crossing Vil1-Cre with LysoTag mice expressing TMEM192-3xHA. Created in BioRender by Eroglu, C., 2025. https://BioRender.com/v929b1h. This figure was sublicensed under CC BY 4.0 terms. (G) Images of P6 and P36 jejunum and ileum, HA-labeled lysosomes. (H) Quantification of lysosomal volume. Lysosome volume is normalized to cell volume (*n*=3 mice/1-5 cells mouse, two-way ANOVA).

Remarkably, in adult animals, relatively few pathways were differentially regulated between regions (Fig. S4G). When we performed gene ontology for biological processes in the jejunum, we found an upregulation of steroid hydroxylase activity (Fig. S4G), which is crucial for the biosynthesis of steroids (Keeney and

Waterman, 1993). In the ileum, we found an upregulation of collagen receptor activity (Fig. S4G), consistent with increased ECM receptor activity in neonates, suggesting ECM activity remains high in the ileum throughout life. When we performed KEGG pathway analysis, we found limited enriched pathways

(Fig. S4H), further supporting reduced regional specialization in adult mice.

Together, these data indicate that regional specialization is higher in neonates than in post-weaning mice. Furthermore, the upregulation of genes involved in endocytic recycling in the neonatal jejunum, compared to the increased expression of proteases in the ileum, suggests differences in their endolysosomal programs. These differences likely underlie the distinct functional roles in protein transcytosis versus degradation of these enterocyte populations.

To further characterize the endolysosomal system, we utilized *LysoTag* mice (Laqtom et al., 2022), which express TMEM192-3xHA specifically in lysosomes upon Cre-mediated recombination. To visualize lysosomes in intestinal epithelial cells, we crossed LysoTag mice with *Vil-Cre* (el Marjou et al., 2004), enabling targeted expression of the HA-tagged lysosomal marker in enterocytes (Fig. 2F). We employed thick vibratome sectioning to preserve the full architecture of enterocytes and performed 3D reconstruction of the lysosomal network within individual cells using Imaris software. In the P6 jejunum, we observed a dense endocytic network with relatively large lysosomes. These findings suggest that TMEM192 not only labels lysosomes but also highlights other components of the endolysosomal system, including endosomes and tubular structures. The density and complexity of this network in the jejunum exceeded the segmentation capabilities of available image analysis tools (Fig. 2G) and likely reflect an elaborate apical endocytic complex required for protein sorting during transcytosis. In contrast, P6 ileal enterocytes typically contained a single large lysosome along with a few smaller vesicles, indicative of a simpler endolysosomal organization. Quantification revealed that endosomes and lysosomes occupied ∼5% of the total cell volume in the jejunum, compared to roughly 17% in the ileum (Fig. 2H). By P36, endolysosomal volume declined to less than 1% in both regions.

Together, these findings demonstrate that although the neonatal jejunum and ileum exhibit high levels of endocytic activity, they differ substantially in their endolysosomal architecture and capacity for intracellular protein digestion. This regional difference likely reflects distinct functional specializations necessary for proper neonatal physiology and metabolism.

### Single-cell analysis reveals divergent neonatal enterocyte programs

While our bulk RNA sequencing of isolated epithelial cells revealed that enterocytes comprised the majority of the input, we could not rule out the contribution of other epithelial cell types to the observed regional gene expression differences. Therefore, we performed single-cell RNA sequencing (scRNA-seq) on epithelial cells isolated from the P6 jejunum and ileum of the same mice. Preprocessing and data analysis were conducted using Seurat (Satija et al., 2015). After applying quality control filters, we retained 30,521 cells (14,719 from jejunum and 15,802 from ileum), which clustered into 14 distinct cell populations (Fig. 3B).

Notably, jejunal and ileal cells largely segregated into distinct areas of the UMAP plot, with limited overlap (Fig. 3A). Using known markers (Jia et al., 2024), we annotated clusters corresponding to enteroendocrine cells (EECs), goblet cells, stem cells and immune cells. In addition, we identified 10 clusters representing enterocytes. Stem cells originated equally from both regions (Table S2). On the other hand, enterocytes clustered into two large lobes – one predominantly composed of jejunal cells (Fig. 3A, top; Fig. 3B, left lobe) and the other of ileal cells (Fig. 3A, bottom; Fig. 3B, right lobe). Ileal cluster 6 included a higher proportion (∼25%) of jejunal cells and was spatially closer to the

jejunal lobe, suggesting a transitional or functionally overlapping enterocyte population between the two regions.

To identify region-specific enterocyte markers and to further investigate differences in endocytic gene expression, we created merged enterocyte clusters for the jejunum and ileum, and performed differential gene expression analysis (Fig. S5A,B). We found that the endocytic genes *Dab2*, *Cubn* and *Amn* were expressed in over 85% of enterocytes overall but at significantly higher levels in the ileum (Fig. 3C). Jejunal enterocytes expressed higher levels of *Paqr9*, *Fam13a*, *Mfsd2a*, *Cd36* and *Serpina1a* – genes that were also among the top upregulated genes in the P6 jejunum from our bulk RNA-seq data (Fig. 3C and Fig. 2C). Notably, these genes were also expressed at higher levels in the P6 jejunum compared to the P36 jejunum. In the ileum, we identified *Cpne8*, *Bex4*, *Ccdc198*, *Cldn8* and *Afp* as unique markers, which similarly showed higher expression in the neonatal ileum compared to adulthood (Fig. 3C).

To further establish functional differences by region, we performed KEGG pathway analysis for genes differentially expressed between jejunal and ileal enterocytes (Fig. 3D). Consistent with bulk RNA-seq findings (Fig. 2D), we found that jejunal enterocytes were enriched for pathways involved in fat digestion and absorption. In the ileum, we observed an upregulation of lysosomal and complement cascade pathways – key components of intracellular protein degradation (Fig. 3D).

To understand the drivers of regional heterogeneity among enterocyte clusters, we performed cluster marker analysis and identified unique gene signatures for each cluster (Fig. S6A,B). KEGG pathway and cellular compartment gene ontology analyses revealed distinct patterns by spatial location within the UMAP (Fig. 3G,H). In the jejunum, clusters positioned closer to the stem cell cluster (clusters 1 and 4) were enriched for ribosomal genes, suggesting they represent earlier-stage or newly differentiated cells (Fig. 3E). Medial clusters (clusters 3 and 4) expressed canonical jejunal markers, while in the ileum, a similar pattern emerged: clusters near the stem cells exhibited ribosomal gene enrichment, while distal clusters (8, 9 and 10) were enriched for pathways related to protein degradation, including vacuolar and lysosomal membrane components (Fig. 3E,F). Although cluster 8 contained relatively few differentially expressed genes, three of the top markers – *Bglap*, *Bglap2* and *Bglap3* – were uniquely expressed in this cluster.

Based on the observation that clusters closer to stem cells expressed higher expression of ribosomal genes, we hypothesized that enterocyte clustering along the UMAP may reflect maturation along the crypt-to-villus axis. Newborn enterocytes migrate from the base of the villus to the tip and are shed within ∼3-4 days (Potten, 1998). This migration is associated with transcriptomic and proteomic changes along the villus tip axis (Harnik et al., 2021; Moor et al., 2018). Moor et al. previously performed laser capture microdissection and bulk RNA-seq from the base to the tip of adult jejunal villi, defining spatial landmark gene signatures (Moor et al., 2018). We labeled these as villus zones 1-5 from base to tip and used them to generate feature plots (Fig. 3G). Expression of lower-zone landmark genes (zones 1 and 2) was enriched in the stem cell and upper UMAP regions, while medial zones (zones 3 and 4) mapped to the central regions, and villus tip markers were restricted to the bottom of the UMAP (Fig. 3H). These data support the idea that transcriptomic clustering is strongly influenced by enterocyte maturation state as it moves from the base to the tip of the villus. Importantly, while the Moor et al. study focused on adult jejunum, we find that these landmark gene expression gradients are conserved in both the neonatal jejunum and ileum, demonstrating the preservation of maturation programs across regions and developmental stages.

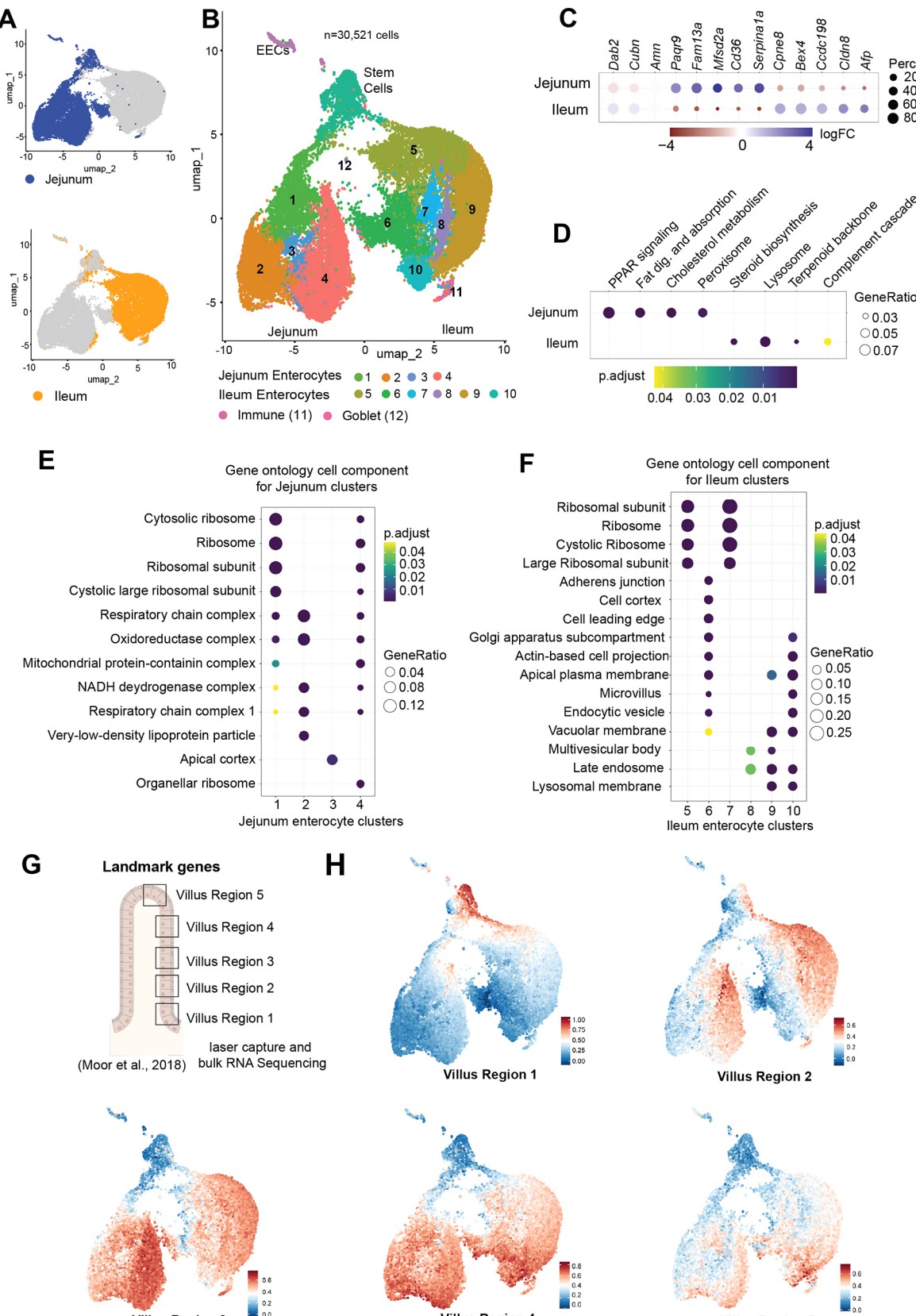

Fig. 3. The neonatal enterocyte program is regionally defined. (A) UMAP showing jejunum (blue) and ileum (orange) clusters. (B) UMAP showing cluster identity and regional specificity of enterocytes. (C) Expression of genes in jejunum and ileum enterocyte clusters. (D) KEGG pathway analysis for differentially expressed genes in jejunum and ileum. (E) Gene ontology for cell components in jejunum enterocyte clusters (1-4). (F) Gene ontology for cell components in ileum enterocyte clusters (5-10). (G) Landmark genes were identified by Moor et al. (2018) by RNA sequencing for five regions of the villus from the base to the tip. (H) Feature plots for landmark genes for each villus region.

### Ileal enterocytes transcriptionally align with zebrafish LREs

Previously, we defined the unique transcriptional signature of LREs in zebrafish, distinguishing them from intestinal epithelial cells (IECs) (Childers et al., 2025). This dataset includes bona fide LREs that were positively identified by fluorescence-activated cell sorting (FACS), with LREs then further validated by their transcriptional signature in downstream scRNA-seq analysis. To assess conservation of the LRE transcriptional program across species, we performed a cross-species analysis integrating this previously published zebrafish scRNA-seq dataset and our novel mouse scRNA-seq datasets. To merge the zebrafish and mouse datasets, zebrafish gene names were converted to mouse orthologs; when no ortholog existed, the zebrafish gene ID was retained. Each zebrafish gene was mapped to a single mouse gene. The zebrafish and mouse datasets were then integrated and analyzed in Seurat (Cortada et al., 2024). The integrated UMAP displayed a similar structure to that of Fig. 3, with mouse jejunal and ileal cells clustering separately (Fig. 4A). Interestingly, zebrafish LREs clustered closely with mouse ileal enterocytes, suggesting a conserved transcriptional identity between zebrafish LREs and the mouse ileum (Fig. 4A).

Consistent with our earlier findings, genes previously identified as enriched in zebrafish LREs continued to show enriched expression in these cells (Fig. 4B). Many of the same genes were also upregulated in mouse ileal enterocytes compared to jejunal enterocytes (Fig. 4B), suggesting that this represents a conserved absorptive enterocyte program. These included the LRE endocytic machinery (*Cubn* and *Dab2*) and lysosomal proteases (*Ctsb*, *Ctsh* and *Scpep1*) (Fig. 4B). Finally, KEGG pathway and GO term analyses revealed that mouse jejunal enterocytes shared enriched pathways with zebrafish IECs, whereas mouse ileal enterocytes shared enriched pathways with zebrafish LREs (Fig. S7A,B), further supporting functional conservation between species and intestinal regions. Zebrafish LREs and the mouse ileum were both significantly enriched in the lysosome KEGG pathway, along with cellular components like late endosomes (Fig. S7A,B). Thus, in line with these findings, we conclude that neonatal enterocytes from the mouse ileum are LREs.

### Human infant enterocytes retain an LRE program that is sharply reduced in NEC

Histological evidence confirmed the presence of LREs in human infants (Fig. 1). However, due to limited tissue availability, it remained unclear whether neonatal enterocytes maintain the molecular machinery required for luminal protein internalization. To address this, we analyzed two published single-cell RNA sequencing datasets that included infant and pediatric ileum (Egozi et al., 2023; Elmentaite et al., 2020).

Computational integration of infant and pediatric enterocytes revealed clear age-dependent segregation (Fig. 4C). Comparing gene expression patterns to those predicted by our mouse RNA-seq analysis revealed that infant enterocytes expressed higher levels of *PRDM1* and *MAMDC4*, key regulators of the neonatal enterocyte program whose loss coincides with the disappearance of vacuolated enterocytes (Fig. 4D) (Bara et al., 2022; Cox et al., 2018; Muncan et al., 2011). Genetic or premature deletion of these genes is known to trigger precocious intestinal maturation (Cox et al., 2018; Muncan et al., 2011). Importantly, *DAB2*, a crucial and neonatal-specific endocytic gene, was expressed in nearly half of infant enterocytes, while other neonatal markers, including *BEX4* and *CD68*, were similarly enriched (Fig. 4D). By contrast, pediatric enterocytes upregulated maturation-associated genes. Housekeeping genes remained stable across ages, confirming meaningful dataset integration (Fig. 4D).

Pathway analysis also showed enrichment in endocytosis, ECM–receptor interactions and PI3K–Akt signaling in infant enterocytes, which are also elevated in neonatal mouse ileum. Pediatric enterocytes upregulated peroxisome and vitamin digestion/absorption pathways (Fig. S8A). Thus, cross-species comparison highlighted conservation of lysosomal and endosomal pathways between infant enterocytes and neonatal mouse ileum, while pediatric enterocytes aligned with post-weaning ileum in their profile (Fig. 4E and Fig. S8B).

To further investigate the infant enterocyte program, infant enterocytes were re-clustered into four groups (Fig. 4F). KEGG pathway analysis revealed an enrichment of lysosomal and endocytic pathways in cluster 0 (Fig. 4G). Consistent with these findings, expression of *DAB2* is enriched and largely overlaps with cluster 0 (Fig. S9A), suggesting a subset of infant enterocytes retains the neonatal endocytic program. Because most available infant intestinal tissue is derived from patients recovering from NEC, we next asked whether NEC impacts the LRE program. Although we observed LREs in NEC patient samples (Fig. 1), these samples were from infants after recovery from NEC, and LREs were less abundant than expected. This raised the possibility that NEC disrupts enterocyte specialization. To explore this, we analyzed infant enterocytes from Egozi et al. (2023), comparing controls (spontaneous ileal perforation, SIP) with NEC. Control samples clustered largely in clusters 0 and 1, whereas in NEC, clusters 0 and 1 was sharply diminished and cells were in clusters 2 and 3 (Fig. S9B,C), which show an enrichment for ribosomal and DNA replication pathways, consistent with accelerated epithelial turnover (Fig. 4G and Fig. S9C).

Together, these findings demonstrate that a subset of infant enterocytes retains a conserved LRE-like neonatal program, including the molecular machinery required for protein uptake. However, this program is lost in NEC, suggesting that disease-associated remodeling leads to depletion of functional LREs.

### Dab2 is required for protein, but not antibody, transcytosis

In zebrafish and mice, loss of Dab2 results in a severe reduction in protein uptake and stunted growth (Park et al., 2019). However, the extent to which Dab2 loss affects LRE development and function in mammals remains unclear. To investigate the impact of Dab2 elimination, we conditionally deleted Dab2 in the intestine by crossing Dab2 floxed mice (Morris et al., 2002) with a Villin-Cre line (el Marjou et al., 2004) (Dab2 cKO) and confirmed protein elimination using immunofluorescence (Fig. 5B). Furthermore, EM in the P6 ileum revealed that Dab2 cKO LREs still harbor a large vacuole (Fig. 5C), consistent with our previous findings that vacuole formation is not Dab2 dependent (Park et al., 2019). However, these vacuoles lacked electron-dense material (Fig. 5C, arrowheads), suggesting that protein uptake is impaired and this defect is already visible at the ultrastructural level. To validate this observation and assess whether Dab2 is required for protein uptake in both the jejunum and ileum, we gavaged neonatal mice with mCherry protein and harvested tissues 3 h later. Quantification of the mCherry signal in the villi revealed a sharp reduction in uptake in both the jejunum and ileum of Dab2 cKO mice (Fig. 5E,F), indicating that Dab2 is essential for non-specific RME in both regions. While the mCherry signal was consistently stronger in the jejunum, this likely reflects differences in endosomal recycling and slower lysosomal degradation, as described in Figs 1 and 2.

Although Dab2 cKO mice exhibit growth stunting, they remain viable into adulthood, suggesting the existence of compensatory metabolic pathways or mechanisms for protein uptake. To investigate this, we performed bulk RNA sequencing on isolated epithelial cells

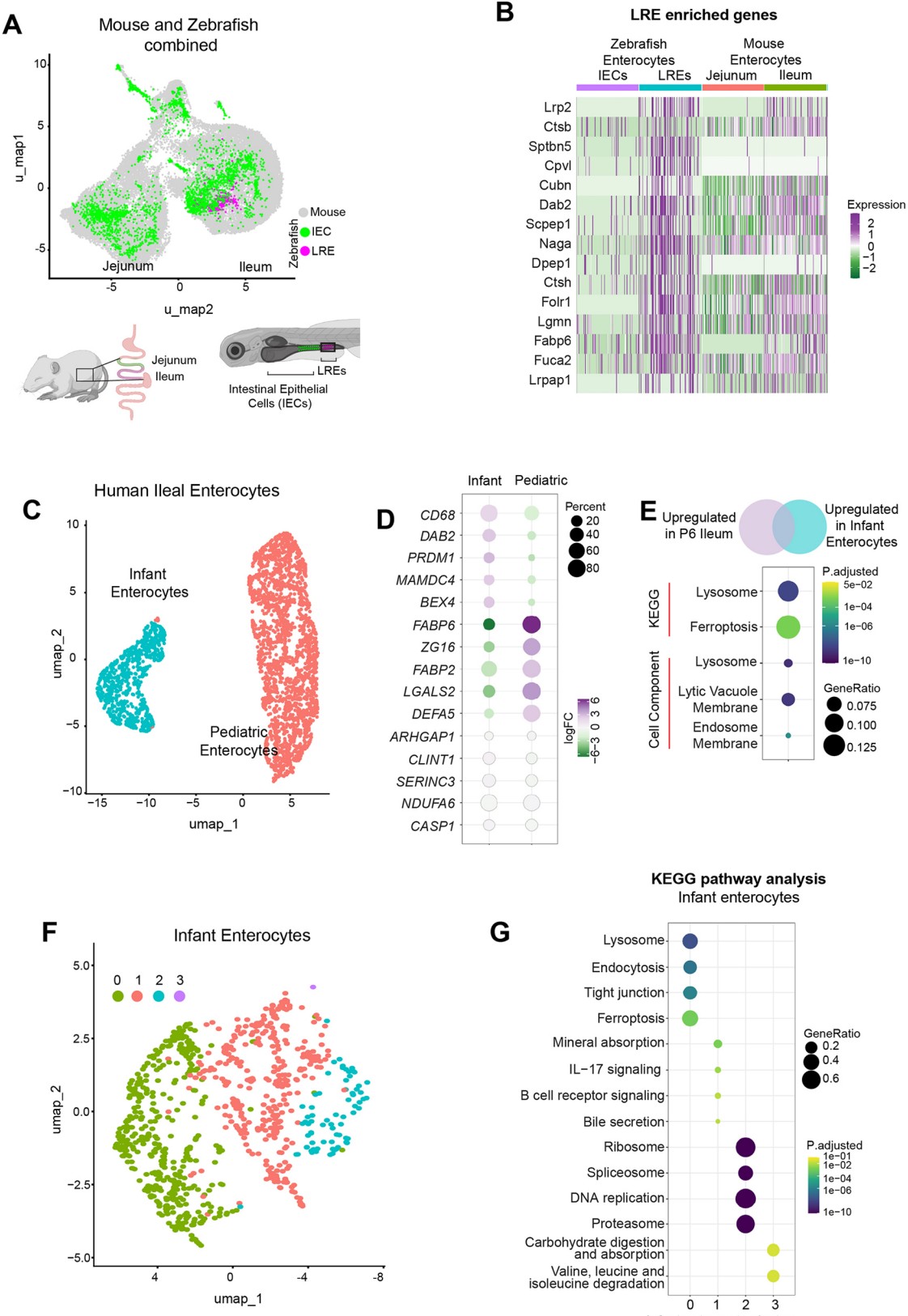

Fig. 4. LRE transcriptional signature is conserved across species. (A) UMAP showing clustering of mouse intestinal cells (gray), zebrafish IECs (green) and zebrafish LREs (magenta). Schematics created in BioRender by Eroglu, C., 2025. https://BioRender.com/0rjb1mp. This figure was sublicensed under CC BY 4.0 terms. (B) Heat map of LRE-enriched genes (Childers et al., 2025) in homologous regions of the zebrafish and mouse intestines. (C) UMAP showing clustering of infant and pediatric enterocytes from Egozi et al. (2023) and Elmentaite et al. (2020). (D) Expression of genes in infant and pediatric enterocyte clusters. (E) Genes upregulated in infant enterocytes and P6 ileum (compared to P36) were compared, and KEGG pathway analysis and gene ontology were performed on overlapping genes. (F) UMAP showing re-clustering of control infant enterocytes. (G) Dot plot of KEGG pathway analysis for infant enterocyte clusters.

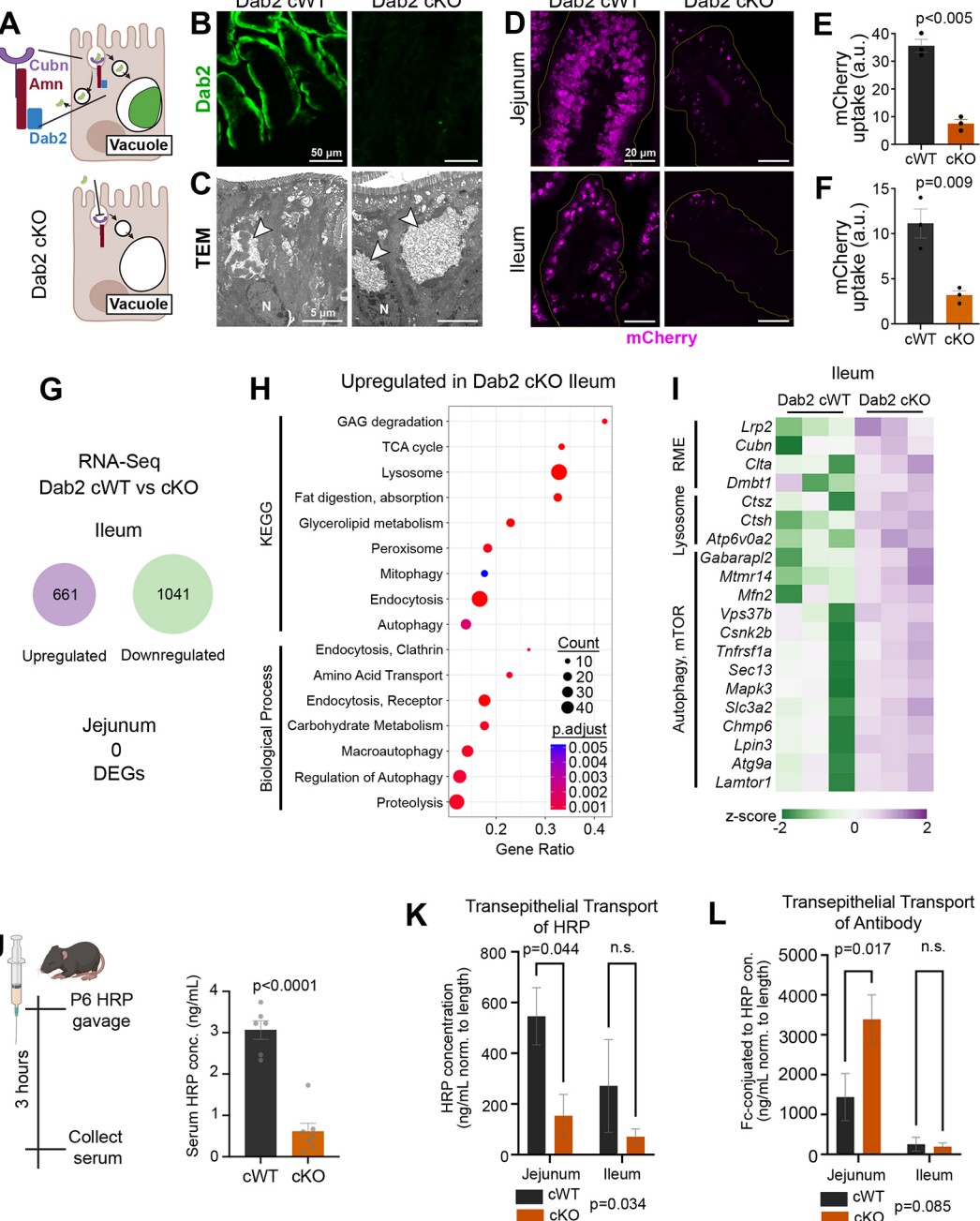

**Fig. 5. Protein uptake is Dab2 dependent in both the jejunum and ileum of the neonatal intestine.** (A) Endocytic machinery is composed of Cubn, Amn and Dab2. Conditional elimination of Dab2 impairs cargo uptake. (B) Immunostaining for Dab2 protein in the ileum of P6 Dab2 cWT and cKO mice. (C) Electron micrographs of the ileum of P6 Dab2 cWT and cKO mice. Arrowheads indicate the presence of a large vacuole. (D) Confocal images of P6 jejunum and ileum showing mCherry signal 3 h post-gavage in Dab2 cWT and cKO mice. (E) Quantification of mCherry signal in the P6 jejunum of Dab2 cWT and cKO mice. mCherry area normalized to villus area (*n*=3 mice/genotype, unpaired *t*-test). (F) Quantification of mCherry signal in the P6 ileum of Dab2 cWT and cKO mice. mCherry area normalized to villus area (*n*=3 mice/genotype, unpaired *t*-test). (G) RNA sequencing from the jejunum and ileum of Dab2 cWT and cKO mice 1741 DEGs in the ileum of Dab2 cKOs (Log$_2$FC>0.58 and FDR<0.25), no DEGs in the jejunum (*n*=3 mice/genotype). (H) KEGG pathways (top) and biological processes (bottom) upregulated in Dab2 cKO ileum. (I) Heatmap of upregulated genes in Dab2 cKO ileum. (J) P6 pups were gavaged with HRP, 3 h post-gavage serum was collected and HRP was measured via ELISA. Quantification of HRP concentration in serum pf P6 Dab2 cWT and cKO mice (*n*= 6 or 7 mice per genotype, unpaired *t*-test). (K) Quantification of HRP concentration in media from an *ex vivo* closed-loop assay in P6 Dab2 cWT and cKO mice (*n*=3 or 4 mice per genotype, two-way ANOVA, post-hoc multiple comparisons). (L) Quantification of antibody transcytosis (mouse anti-human IgG-HRP) in the jejunum of P6 Dab2 cWT and cKO mice (*n*=3 or 4 mice per genotype, two-way ANOVA, post-hoc multiple comparisons). Panels A and J created in BioRender by Eroglu, C., 2025. https://BioRender.com/j3w19r8. This figure was sublicensed under CC BY 4.0 terms.

from the jejunum and ileum of Dab2 cKO mice. In the ileum, we identified 1741 differentially expressed genes, whereas no significant gene expression changes were observed in the jejunum between cWT (conditional wild-type) and Dab2 cKO mice (Fig. 5G). These results

suggest that the ileum is more dependent on Dab2 and undergoes a more robust transcriptional response to compensate for its loss.

Gene ontology analysis of the DEGs from cWT and Dab2 cKO ileum revealed downregulation of genes involved in the cell cycle

and mitosis, consistent with a starvation-induced growth arrest (Fig. S10). At the same time, we observed upregulation of alternative endocytic and nutrient-scavenging pathways. Notably, expression of the scavenger receptor megalin (*Lrp2*) – a key player in RME in the kidney – was elevated, even though it is typically expressed at low levels in LREs (Nielsen et al., 2016). Additionally, clathrin light chain (*Clta*) expression was increased, suggesting a compensatory upregulation of canonical clathrin-mediated endocytosis (Fig. 5H,I). We also observed an enrichment of lysosome biogenesis, autophagy and macroautophagy-related pathways, in DEG GO terms, further supporting a cellular response to nutrient deprivation (Fig. 5H,I). Taken together, these findings suggest that the ileum is not only the primary site of protein digestion and absorption but may also rely on nutrients generated through the intracellular digestion of luminal proteins for its own metabolic needs.

Next, we tested whether Dab2 is required for the uptake of proteins destined for transepithelial transport into the neonatal bloodstream. To test this, we gavaged P6 Dab2 cWT and cKO mice with HRP and measured its levels in serum 3 h later. Circulating HRP was significantly reduced in Dab2 cKO mice, indicating that impaired Dab2-mediated protein uptake is needed for protein transcytosis (Fig. 5J).

Because the jejunum is the principal site of trans-epithelial transport (Fig. 1K,L), we used our *ex vivo* closed-loop assay to test whether jejunal transcytosis is Dab2 dependent. We found that Dab2 deletion significantly reduced HRP transcytosis in the jejunum (Fig. 5K), reinforcing its central role in systemic protein transport.

While HRP utilizes non-specific protein uptake machinery, maternal IgG is transcytosed via FcRn-mediated endocytosis. To determine whether Dab2 is involved in this process, we conducted the same closed-loop assay, this time using HRP-conjugated antibodies. Dab2 cKO did not impair antibody transcytosis in the jejunum, but resulted in a significant increase (Fig. 5L). These findings demonstrate that Dab2 is not required for FcRn-mediated endocytosis and support the existence of distinct endocytic mechanisms, such as clathrin-mediated endocytosis, for nutrient-derived proteins and maternal antibodies. Additionally, our study indicates that these alternative endocytic pathways are utilized as compensatory mechanisms upon nutrient deprivation when Dab2 is lost.

## DISCUSSION
While the division of labor within the neonatal intestine has been previously hypothesized, it had not been directly shown due to the lack of quantitative assays and approaches for dissecting the relative contributions of each intestinal region. Using *in vivo* and *ex vivo* assays, we show that, although both jejunum and ileum are capable of endocytosing intact proteins, the jejunum preferentially mediates their transepithelial transport, while the ileum favors lysosomal degradation. Previous work reported that growth factors EGF and NGF undergo transepithelial transport in the ileum but are substantially degraded (Gonnella et al., 1987; Siminoski et al., 1986), leaving unclear the relative transport capacity of intestinal segments. Our data show that while the jejunum has a much higher capacity for transepithelial transport, a small amount of transport is also taking place in the ileum. This functional distinction is reflected in both functional assays and transcriptomic signatures. Bulk and single-cell RNA sequencing reveal that neonatal jejunal enterocytes upregulate genes associated with lipid metabolism, transcytosis and endocytic recycling, whereas ileal enterocytes express genes linked to lysosomal function, proteolysis and complement activation. Thus, our study redefines the functional and molecular landscape of

neonatal enterocytes in the small intestine, revealing divergent modes of protein processing: transcytosis in the jejunum and intracellular degradation in the ileum.

Remarkably, regional specialization of the small intestine is most pronounced during early stages of development. At P6, the number of DEGs between the jejunum and ileum is more than double that observed at P36. Many of these genes are uniquely expressed in neonates; furthermore, some of the top DEGs by region at P6 – such as *Fam13a*, *Paqr9*, *Rarres1* and *Cpne8* – have not been previously characterized in the intestinal epithelium, indicating that crucial aspects of neonatal jejunal and ileal function remain poorly understood. Our findings reveal that the neonatal intestine is not simply a scaled-down version of the adult gut, but rather a uniquely adapted system optimized to mediate passive immunity and meet the metabolic demands of rapid growth. Moreover, our transcriptomic data suggest there are likely other unknown functions that will be important to explore in future studies.

Our single-cell analysis further highlights regional divergence in neonates. While the jejunum and ileum contributed equally to the stem cell population, this population appeared to give rise to two major classes of enterocytes – one predominantly composed of jejunal cells and the other of ileal cells (Fig. 4A,B). These data suggest the existence of region-specific differentiation programs regulated by unknown upstream cues.

In our single-cell analysis, both jejunal and ileal enterocytes further clustered within each class, likely reflecting migration along the villus axis (Fig. 3). However, we did identify subclusters with unique gene expression patterns, such as the ileal enterocyte cluster 8 (Fig. S6B), which uniquely express *Bglap*, *Bglap2* and *Bglap3*. These genes encode osteocalcins that are primarily expressed by osteoblasts and are involved in regulating bone calcification, and more recently have been known to mediate endocrine function (Mizokami et al., 2017). Osteocalcin was also found upregulated in the gut of *Smarcad1*-KO, but its function in the intestinal epithelium remains poorly understood (Kazakevych et al., 2020).

Our cross-species comparisons with zebrafish further support the evolutionary conservation of regionally specialized enterocyte programs. Zebrafish LREs transcriptionally cluster with mouse ileal enterocytes and share enrichment in lysosomal and protein degradation pathways, whereas mouse jejunal enterocytes align with zebrafish IECs enriched in lipid metabolism. These data suggest that the division of labor between transcytosis and degradation is a conserved feature of vertebrate intestinal development.

The adaptor protein Dab2, a crucial component of the Cubilin/Amnionless scavenger receptor complex, emerges as a central regulator of non-specific protein uptake. Conditional deletion of Dab2 significantly impairs protein internalization in both the jejunum and ileum, yet only the ileum mounts a robust compensatory response. Although the jejunum and ileum both utilize the scavenger receptor complex for the non-specific uptake of luminal proteins, the jejunum also expresses a robust and dedicated machinery for the transport of maternal antibodies.

Importantly, Dab2 cKO mice still retain neonatal enterocyte morphology, including the presence of a large lysosomal vacuole. This phenotype contrasts with other genetic models, such as the endotubin (EDTB) knockout – encoded by the *Mamdc4* gene – in which endotubin is essential for the biogenesis of the apical endocytic complex and the formation of the large lysosomal vacuole (Cox et al., 2018; Wilson et al., 1991). Notably, Dab2 deletion does not significantly alter *Mamdc4* expression; however, both the neonatal jejunum and ileum exhibit a ~10-11 $\log_2$fold increase in *Mamdc4* expression compared to P36, consistent with neonatal-specific

features that rely on *Madmdc4*, such as the presence of large lysosomes and a robust apical endocytic system (Cox et al., 2018). Transcriptomic profiling of Dab2 cKO mice reveals upregulation of alternative endocytic and lysosomal pathways in the ileum but not the jejunum, underscoring regional differences in metabolic reliance on luminal protein degradation. Importantly, *Dab2* is dispensable for FcRn-mediated antibody transcytosis, supporting the notion that nutrient-derived proteins and maternal immunoglobulins follow distinct intracellular routes. In line with this, we found that the neonatal jejunum is substantially more efficient at antibody transcytosis compared to HRP (Fig. 5K,L), likely due to the presence of a dedicated receptor (FcRn) for immunoglobulin transcytosis.

Our findings also challenge the long-held assumption that LREs are absent in humans after birth. Through PAS staining and electron microscopy, we identify LREs in the ileum of human neonates up to 4 months of age, with features intermediate between neonatal and post-weaning mouse LREs. This finding expands the developmental window during which LREs are functionally relevant in human infants and suggests that their contributions to neonatal nutrition and immune programming may be underappreciated.

Analysis of previously published single-cell RNA sequencing further revealed that infant enterocytes retain a conserved LRE-like neonatal program, including *PRDM1*, *MAMDC4* and *DAB2*. Notably, this program was disrupted in NEC, where *DAB2*-positive clusters were selectively lost, and NEC enterocytes instead showed signatures of accelerated epithelial turnover. Together, these findings establish that infant enterocytes maintain an LRE program that is developmentally regulated but vulnerable to pathological disruption in NEC.

## Conclusions

This work provides new evidence illustrating how the neonatal intestine compartmentalizes protein handling along the proximal-distal axis. By combining structural, functional and molecular analyses, we reveal that jejunal and ileal enterocytes are uniquely specialized for protein transcytosis and degradation, respectively. Our findings underscore the crucial role of the Dab2-mediated endocytic machinery in supporting these functions, highlighting the dynamic nature of epithelial specialization during early life. Importantly, the presence of LREs in human neonates points to a conserved role for these cells in mammalian development. These insights have broad implications for neonatal nutrition, passive immunity and disease susceptibility, particularly in the context of malabsorption syndromes, protein-losing enteropathies and the care of premature infants. Future studies should explore how these pathways are regulated by diet, microbiota and inflammation, and how their dysregulation contributes to pediatric gastrointestinal disorders.

## MATERIALS AND METHODS
### Experimental animal models

All experiments were performed with Institutional Animal Care and Use Committee approval from Duke University, and in accordance with the guidelines of the Division of Laboratory Animal Resources guidelines for the humane treatment of animals. Mouse lines used in this study were: wild type (C57BL/6J; JAX:000664, RRID:IMSR_JAX:000664), Villin-Cre [B6N.Cg-Tg(Vil1-cre)20Syr/J, JAX:033019, RRID: IMSR_JAX:033019] (el Marjou et al., 2004), Lysotag [B6.129S4-Gt(ROSA)26Sortm1(CAG-TMEM192)Dmsa/J; JAX:035401; RRID: IMSR_JAX:033019] (Laqtom et al., 2022) and Dab2 fl/fl (B6;129S4-*Dab2^{tm1Cpr}*/J; JAX:022837; RRID: IMSR_JAX:022837) (Morris et al., 2002). Both male and female mice were used. Mice were maintained in a facility with a 12 h light/dark cycle with *ad libitum* access to food and water.

Histology, bulk-RNA sequencing and lysosome reconstructions were performed in P6 and P36 mice. The *ex vivo* uptake assay was performed in P3 mice. All other assays, including *in vivo* uptake and degradation assays, *ex vivo* transport assays, single-cell RNA sequencing and all Dab2 cKO experiments, were performed in P6 mice.

### Regional dissection

For all experiments in mice, the intestines were removed, unraveled and the mesentery was removed. The intestines were then segmented into three equal parts representing the duodenum, jejunum and ileum. Since the small intestine does not have strong boundaries, these sections were further divided into three equal parts, and the medial region was used to represent the jejunum and ileum.

### Human patient samples

Intestinal samples (distal ileum) were obtained from human neonates with a history of intestinal resection for necrotizing enterocolitis undergoing surgery for ileostomy takedown and reanastomosis (*n*=2 with gestational ages of 23 and 28 weeks with postmenstrual ages at the time of surgery of 35 weeks and 58 weeks, respectively). All human tissue was obtained and processed with parental consent and approval from the University of North Carolina at Chapel Hill Institutional Review Board (Protocols 21-3134 and 21-2846), and in accordance with the University of North Carolina at Chapel Hill anatomical tissue procurement guidelines.

### Soluble fluorescent protein production

mCherry and mTurquoise soluble fluorescent proteins were produced by uninduced expression in the BL21-Gold (DE3) strain of *Escherichia coli*. The fluorescent proteins were purified from the pellet of large bacterial culture using the His tag (Sarabipour et al., 2014).

### Imaging
#### Histology

For histological analysis, mice were euthanized via rapid decapitation for neonates or carbon dioxide gas for weanlings. Intestines were removed, washed in PBS, and placed into 4% PFA for overnight fixation. For the human patient samples, fixation was in EM buffer (detailed below) for 1 week, and the remaining steps were identical. Tissue was washed and embedded in 1% agar in water and dehydrated in ethanol. Dehydrated tissue was placed into JB4 infiltration solution overnight, then placed into plastic resin embedding solution (JB4-plus kit, 18570C, Polysciences). Plastic blocks were cut at 1-2 µm using a glass knife on a microtome (Leica, RM2265). Plastic sections were stained with Periodic Acid-Schiff (395B-1KT, Sigma-Aldrich) and counterstained using Hematoxylin solution, gill no. 3 (395B-1KT, Sigma-Aldrich), coverslipped with Cytoseal XYL (8312-4, Epredia) and imaged on an Axioimager Z1 (Zeiss) at 100× magnification. Three images were acquired per animal. Each image was analyzed by a researcher who was unaware of the condition, age and sex. Within a field of view, the number of epithelial cells along the villus was counted, including enterocytes and goblet cells (bright magenta). Vacuolated cells were defined by the presence of a large clear vacuole in their cytoplasm. Vacuolates could contain PAS-positive material, but were clearly distinguishable from the cytoplasm. The percentage of vacuolated cells was calculated over the total number of counted villus epithelial cells.

#### Electron microscopy

Mice were euthanized via rapid decapitation, intestines were removed, washed in PBS and fixed overnight in EM buffer [2% PFA (Pierce, 28908) and 2% glutaraldehyde (Electron Microscopy Sciences, 16320) in 0.1 M sodium cacodylate buffer (Electron Microscopy Sciences, 11650)]. For all specimens, preparation was carried out as previously described (Ding et al., 2015). In short, after staining with osmium tetroxide, samples were dehydrated in ethanol solutions of increasing concentration, embedded in resin blocks overnight, and then embedded and cured in a 60°C oven for 48 h. Thin sections were then cut and post-stained with lead citrate and uranyl acetate, and placed on copper grids for imaging. Samples were imaged on a JEM-1400 electron microscope (JEOL) at 60 kV with a digital camera (BioSprint; AMT).

## In vivo gavage assays

### Uptake assays and immunofluorescence

For *in vivo* uptake assays, P6 mice were gavaged with 20 µg mCherry in a total volume of 10 µl, with a plastic feeding tube (Instech, FTP-22-25-50). Mice were euthanized, and the intestines were collected 3 h post-gavage. The intestine was collected and placed into 4% PFA overnight. Control assays were performed with PBS gavage in P6 mice or mCherry gavage in a P36 mouse (50 µg mCherry in a volume of 25 µl, FTP-22-38-50). For uptake assays in Dab2 cWT and cKO mice, littermate pairs were used. The fixed intestines were washed and then transferred to 30% sucrose for cryoprotection. Samples were frozen and embedded in OCT (Tissue Tek, Sakura, 25608-930) and stored at −80°C. Intestines were sectioned at 12 µm, thaw mounted on Superfrost Plus slides (VWR, 48311-703) and stored at −20°C until use.

Slides were acclimated to ambient temperature and then washed three times with PBS. A hydrophobic barrier was generated with a PAP pen (Diagnostic Biosystems, K039) and slides were incubated in antibody diluent (Dako, S0808) and phalloidin (1:100, Invitrogen, A12379) for 1 h at room temperature, and then incubated with DAPI (1:100,000, Invitrogen, D1306) in PBS for 5 min. Slides were washed three times in PBS, then coverslipped with an in-house mounting media (20 mM Tris pH 8.0, 90% glycerol and 0.5% N-propyl gallate). Images were acquired using an Olympus FV3000 laser scanning confocal at 60×, 3 µm z-stacks were acquired using a 0.33 µm step size. Images were max projected, blinded and thresholding for mCherry was performed using Fiji (1.54p).

### Degradation assays and immunofluorescence

For *in vivo* degradation assays, P6 mice were gavaged with 20 µg mCherry and 600 µg of mTurquoise in a total volume of 30 µl with a plastic feeding tube (Instech, FRTP-22-25-50). Mice were collected either 3 h post-gavage or 5 h. Intestines were collected and placed into 4% PFA overnight. The fixed intestines were washed and then transferred to 30% sucrose for cryoprotection. Samples were frozen and embedded in OCT (Tissue Tek, Sakura, 25608-930) and stored at −80°C. The intestines were sectioned at 12 µm and then thaw-mounted on Superfrost Plus slides (VWR, 48311-703) and stored at −20°C until use.

Slides were acclimated to ambient temperature and then washed three times with PBS, then coverslipped with an in-house mounting media [20 mM Tris (pH 8.0), 90% glycerol and 0.5% N-propyl gallate]. Images were acquired using an Olympus FV3000 laser scanning confocal at 60×, 3 µm z-stacks were acquired using a 0.33 µm step size. Images were max projected and anonymized, and thresholding for mCherry was performed using Fiji (1.54p).

### Transcytosis assays

For the *in vivo* transcytosis assay, P6 Dab2 cWT and cKO littermate pair mice were gavaged with 15 µg of HRP per gram of weight (~3-4 µl total volume, HRP concentration 15 µg/µl). At 3 h post-gavage, mice were euthanized via rapid decapitation and trunk blood was collected into a sterile 1.7 ml tube. Blood was allowed to coagulate for 30 min at room temperature. To separate serum from red blood cells, blood was centrifuged twice at 21,130 *g* for 10 min and was transferred into a sterile tube and stored at −20°C until analysis. For analysis, a standard curve was generated, 10 µl of sample was loaded into a 96-well plate, 40 µl of TMB solution was added and, after 30 min, 40 µl of Stop buffer was added. Absorbance was measured at 450 nm on a Spectra Max M5 plate reader (Molecular Devices). ELISA reagents were obtained from the ELISA kit (Invitrogen, CNB0011). Standards (10 µl) were pipetted into duplicate into wells with 10 µl of sample.

## Ex vivo closed-loop assays

### Protein uptake and immunofluorescence

For protein uptake assays, P3 mice were euthanized via rapid decap. Intestines were removed and placed into HBSS with calcium and magnesium (HBSS++, Gibco, 14040133). The mesentery was gently removed, the intestine was cut into three equal parts, and the intestinal lumen of the jejunum and ileum was gently flushed with HBSS++ and placed into intestinal media [50% DMEM-F12 (Gibco, 21041025) with 1% Glutamax (Gibco, 21041025), 1 mM penicillin-streptomycin (Gibco, 15140-122), 10 mM HEPES (Gibco, 15630080), 1% N2 (Gibco, 17502048), 1 µM NAC

(Sigma, A8199) and 50% L-WRN (Sigma, SCM105)] for 5 min. After acclimation, the lumen was filled with 50 µg of mCherry in a volume of 25 µl and the lumen was tied off with sewing thread. Samples were incubated in intestinal media in an incubator at 37°C (5% $CO_2$ at 37°C) for 30 min, then flushed with HBSS and placed into 4% PFA overnight at 4°C. Fixed intestines were washed and transferred to 30% sucrose for cryoprotection. Samples were frozen and embedded in OCT (Tissue Tek, Sakura, 25608-930) and stored at −80°C. Intestines were sectioned at 12 µm, thaw mounted on Superfrost Plus slides (VWR, 48311-703) and stored at −20°C until use.

Slides were acclimated to ambient temperature and then washed three times with PBS. A hydrophobic barrier was generated with a PAP pen (Diagnostic Biosystems, K039) and slides were incubated in antibody diluent (Dako, S0808) and phalloidin (1:100, Invitrogen, A12379) for 1 h at room temperature, and then incubated with DAPI (1:100,000, Invitrogen, D1306) in PBS for 5 min. Slides were washed three times in PBS, then coverslipped with an in-house mounting media [20 mM Tris (pH 8.0), 90% glycerol and 0.5% N-propyl gallate]. Images were acquired using an Olympus FV3000 laser scanning confocal at 60×, 3 µm z-stacks were acquired using a 0.33 µm step size. Images were max projected and anonymized, and thresholding for mCherry was performed using Fiji (1.54p).

### Transcytosis

For transcytosis assays, P6 mice were euthanized via rapid decapitation. For transcytosis in Dab2 cWT versus cKO mice, littermate pairs were used. Intestines were removed and placed into HBSS with calcium and magnesium (HBSS++, Gibco, 14040133). The mesentery was gently removed, the intestine was cut into three equal parts, the intestinal lumen of the jejunum and ileum was gently flushed with HBSS++, and placed into intestinal media [DMEM-F12 with 1% Glutamax, 1 mM penicillin-streptomycin, 10 mM HEPES, 1% N2, 1 µM NAC and 2% B27 (Gibco, 17504044)] for 5 min. After acclimation, the lumen was filled with 30 µg HRP (Sigma-Aldrich, P8250), 40 mM biotin (Sigma-Aldrich, B4501) and 10 mg/ml AlexaDye (Invitrogen, A33077), or with 20 µg of IgG conjugated to HRP (Invitrogen, 31420) in a volume of 30 µl. Each end of the lumen was ligated with sewing thread, and samples were washed twice in 25 ml DPS to remove any residual protein. Samples were incubated in 3 ml of intestinal media in a 12-well plate and incubated at 37°C (5% $CO_2$ at 37°C) for 60 min, 150 µl of media was collected into a sterile Eppendorf tube at time 0, 15, 30 and 60 mins.

The concentration of each cargo in the collected media was measured as follows. For HRP or IgG conjugated to HRP, samples were diluted at 1:100 in ELISA assay buffer prepared alongside a standard curve with a similar dilution. Samples were loaded in triplicate into a 96-well ELISA plate (10 µl). 100 µl of TMB solution was added and, after 30 min, 100 µl of Stop buffer was added. Absorbance was measured at 450 nm on a Spectra Max M5 (Molecular Devices) plate reader. ELISA reagents were obtained from the ELISA kit (Invitrogen, CNB0011). All sample concentrations were normalized to the length of the sample. For Biotin, the Pierce Fluorescence Biotin Quantitation Kit (46610) was used, using a DyLight reporter. Samples were loaded in duplicate, alongside a standard curve (10 µl of sample). Fluorescent excitation/emission 494/520 nm were measured using a Spectra Max M5 (Molecular Devices) fluorescent microplate reader. All sample concentrations were normalized to the length of the sample. For AlexaDye, 2 µl of undiluted sample was loaded onto a nanodrop in quadruplicate, and the concentration of AlexaDye 488 was measured. All sample concentrations were normalized to the length of the sample.

## Cell isolation and bulk RNA sequencing

### Epithelial cell isolations

Mice were euthanized via rapid decapitation for neonates or carbon dioxide gas for weanlings. Intestines were removed, placed into PBS and the mesentery was gently removed. The intestine was cut into three equal parts, and a 1 cm segment of the center of the jejunum and ileum was retained. Samples were cut open longitudinally, washed in PBS and gently agitated to remove any luminal contents. Sections were incubated in 5 ml of 30 mM EDTA in PBS. Intestines from P6 mice were incubated at 37°C for 10 min with rotation and intestines from P36 mice were incubated for 10-15 min at

37°C with rotation. The intestines were transferred to a Petri dish with PBS, grasped with forceps and then shaken to release the epithelium. The epithelial sheets were transferred to a 15 ml conical tube and allowed to pellet by gravity, and were washed twice with PBS. For bulk RNA sequencing, samples were transferred to a 1.7 ml tube, snap frozen and stored at −80°C until RNA extraction. For single-cell analysis, epithelial sheets were immediately processed to generate a single-cell suspension.

### RNA extraction
Frozen samples were homogenized in 1000 μl TRIzol Reagent (Invitrogen, 15596026) and vortexed on a MixMate at 2000 rpm for 5 min. 200 μl of chloroform (Sigma-Aldrich, C2432) was added to each tube and vortexed for an additional 2 min; samples were phase separated before being centrifuged at 13,523 $g$ for 15 min at 4°C, after which the top clear aqueous phase was separated into a fresh tube. 500 μl of isopropanol was added, samples were vortexed on a MixMate at 2000 rpm for 1 min and incubated at room temperature for an additional 10 min, and then centrifuged for 10 min. The supernatant was discarded, and the RNA pellet was washed twice with 1 ml of ice-cold 75% ethanol. It was then air-dried and resuspended in 40 μl of RNase-free water. DNA contaminants were removed using the RNA Clean & Concentrator-5 kit according to the manufacturer's recommendations (Zymo, R1014). Purified samples were stored at −80°C until ready for shipment. All samples were blinded and shipped to Medgenome for library prep and sequencing using the Kapa Stranded mRNA-seq kit and sequenced on a NovaSeq 6000. Raw reads were adapter trimmed using Trimmomatic (v0.38) (Bolger et al., 2014), aligned to the reference mouse genome (mm10; GRCm38) using STAR (v2.3.5a) (Dobin et al., 2013) and counted using Subread (featureCounts, v1.6.3) (Liao et al., 2013). Differential gene expression was conducted using edgeR (v4.6.3) (Robinson et al., 2010). Gene-Ontology terms (GO terms) and KEGG pathway analyses were carried out with R using the clusterProfiler (v4.17.0) package (Yu et al., 2012). Sequencing results for developmental analysis have been deposited in GEO under accession number GSE301314 and can also be explored with a user-friendly interface at our website: https://eroglulab.shinyapps.io/Carina_GutRNA/. Sequencing results for Dab2 cWT vs cKO animals have been deposited in GEO under accession number GSE301313.

### Lysosome reconstructions
For lysosome reconstructions, Vil1-Cre mice (el Marjou et al., 2004) were crossed with Lysotag mice (Laqtom et al., 2022), and mice positive for cre and with one transgene for lysotag were collected. P6 mice were euthanized via rapid decapitation, and the intestines were removed and placed into PBS to remove the mesentery. A 2 cm medial segment of the jejunum and ileum was removed and placed into 4% PFA. The intestines were fixed for 4 h at room temperature and then washed three times in PBS. Intestinal segments were embedded into 5% low-melt agarose in PBS and sectioned at 125 μm on a vibratome (Leica, VT1000). For staining, sections were washed three times in PBS, incubated for 1 h in blocking buffer (5% NGS and 3% BSA in 0.05% Triton-X), then incubated in primary antibody overnight at 4°C in blocking buffer (1:200, anti-HA, Roche, 12352203). The next day, sections were washed three times in PBS, and transferred to secondary antibody staining in blocking buffer for 2 h (phalloidin, 1:100, Invitrogen, A12379; goat anti-rat 594 1:500, Invitrogen, A11007). During the last 10 min of secondary incubation, DAPI (Invitrogen, D1306) was added at a concentration of 1:100,000. Sections were then washed three times in PBS, mounted and coverslipped with an in-house mounting media [20 mM Tris (pH 8.0), 90% glycerol and 0.5% N-propyl gallate). High-resolution imaging was performed on an Olympus FV3000 confocal laser-scanning microscope with a 60× objective with 1.5 zoom at a step size of 0.32 μm. Huygens Professional 19.10.0p3 64b was used to deconvolve images. Imaris 9.5.1 was used to create surface renderings, by manually drawing surfaces based on HA staining.

### Single-cell RNA sequencing
The medial jejunum and ileum of a P6 male and female mice were isolated and rinsed in PBS. The tissue was segmented into ~2 cm fragments and opened longitudinally to expose the epithelium. The tissue was incubated in 30 mM EDTA in HBSS (Gibco, 14170112) at 37°C for 15 min. The tissue was vigorously shaken to release epithelium. The epithelium was collected into a 15 ml conical tube, incubated on ice and allowed to settle via gravity. The epithelium was washed twice in HBSS, then transferred to a C-tube with 1 Unit/ml dispase II (Gibco, 17105041) and 20 mg/ml of DNase (Worthington, LS002007) in HBSS and titurated with P1000 for 10 passes before being incubated in a gentleMAC dissociator for 12 min at 200 rpm at 37°C. Cells were titurated for 10 more passes with a P1000 pipette, then incubated for another 12 min at 200 rpm at 37°C. Cells were passed through a 100 μm strainer in 10% FBS (Sigma-Aldrich, F2442) and collected by centrifugation at 300 $g$ for 3 min at 4°C. Cell pellets were washed in cell suspension buffer (from the PIPSeq kit) and centrifuged again at 300 $g$ for 3 min at 4°C. Cells were resuspended in cell suspension buffer and run through a 30 μm filter. Viability was counted using green/red viability stain (Invitrogen, A49905) on a Countess3 FL (Invitrogen). All cell viabilities were above 95% and cells were diluted to a cell concentration of ~4000 cells/μl. Cells were then run through particle-templated instant partition sequencing according to the manufacturer's recommendations (Fluent Bioscience, PIPseq V T20 3′ Single RNA Kit) (Clark et al., 2023). Sequencing of the libraries was performed by Medgenome on NovaSeq (PE100) for 625 million reads per library (~30,000 reads per cell). Sequencing results have been deposited in GEO under accession number GSE301381.

Reads were aligned using the pipseeker algorithm at a resolution of 3 and then read into Seurat (v5.3) (Hao et al., 2024). Reads were filtered by the following criteria: minimum cells=3, minimum features=200, genes per cell=1000-6000 and percent mitochondria <0.2. After samples were merged into a single Seurat object, data were normalized and scaled, and linear dimensional reduction was performed with principal component analysis. Layers were integrated using the CCAIntegration method. Cells were clustered with a resolution of 1. Cluster markers were identified using Seurat's 'FindMarkers' function. Cluster identity was determined by cross-referencing marker genes to other scRNA-seq and RNA-sequencing datasets.

### Cross-species comparison
Previously published zebrafish single-cell data from conventionalized zebrafish larvae were integrated with neonatal mouse data (Childers et al., 2025). Zebrafish gene names were converted into homologous mouse gene names using the gProfiler webtool (https://biit.cs.ut.ee/gprofiler/orth) (Kolberg et al., 2023). If zebrafish genes mapped to more than one mouse homolog, the first mouse gene on the list was selected. If no mouse homolog was found, the zebrafish gene ID was retained. This converted features list, and matrix and barcode files from the conventionalized (CV) zebrafish datasets were read into Seurat. Reads were filtered: minimum cells=3, minimum features=200, genes per cell=200-4000, percent mitochondria <0.4. The Seurat object for cross-species analysis was generated by merging the mouse and zebrafish Seurat objects, then integrating them using the CCAIntegration method. Cells were clustered with a resolution of 0.5.

### Human single-cell analysis
Raw count matrices were downloaded from Egozi et al. (2023) and Elmentaite et al. (2020). Pre-processing was performed as recommended by Egozi et al. (2023). Briefly, the average UMI count for each gene in cells identified as background (100-300 UMIs and mitochondrial fraction below 50%) was subtracted from the respective counts in all other cells. Cells with fewer than 200 genes and genes expressed in fewer than three cells were removed from the data.

All samples, including infant control, infant NEC, pediatric control and pediatric Crohn's, were merged. For the comparison between infant and pediatric samples, only control samples were included. Annotation included by the authors was used to subset only enterocytes in both control infants and pediatric samples. For the comparison between infant control and NEC samples, only enterocytes were included in the analysis. Cells with less than 1900 UMIs, less than 1000 genes per cell or a mitochondrial fraction above 30% were filtered out. In addition, cells with a fraction of erythrocyte markers above 10% were filtered out. Cells were normalized and scaled using the SCTransform function, with regression of the sum of UMIs. For the comparison between infant and pediatric samples, principal component analysis (PCA) was calculated based on the variable genes (excluding mitochondrial and ribosomal genes). For the infant samples, PCA was calculated based on all genes (excluding mitochondrial and ribosomal genes). Cells were clustered with a resolution of 0.2. To determine the infant

versus pediatric program, enterocytes from infants and pediatric patients were renamed to form a single cluster per age. Cluster markers were identified using Seurat's 'FindMarkers' function and enrichr was used to perform KEGG pathway analysis (Hao et al., 2024; Chen et al., 2013; Kuleshov et al., 2016; Xie et al., 2021).

## Validation of the Dab2 knockout
### Sample preparation
Dab2 cWT and cKO littermate pair mice were euthanized via rapid decapitation at P6, and the intestines were collected and placed into 4% PFA overnight. Fixed intestines were washed and transferred to 30% sucrose for cryoprotection. Samples were frozen and embedded in OCT (Tissue Tek, Sakura, 25608-930) and stored at −80°C. Intestines were sectioned at 12 μm and thaw mounted on Superfrost Plus slides (VWR, 48311-703) and stored at −20°C until use.

### Immunofluorescence
Slides were acclimated to ambient temperature and then washed three times with PBS. A hydrophobic barrier was generated with a PAP pen (Diagnostic Biosystems, K039) and slides were incubated in protein block (Dako, X0909) for 30 min at room temperature. The primary antibody was prepared in antibody diluent with background reducing agent (Dako, S3022) at a concentration of 1:100 (Dab2, BD Biosciences, 61046) and incubated at 4°C overnight. Slides were washed three times in PBS, and then incubated in secondary antibody in antibody diluent buffer (Dako, S0808) for 1 h at room temperature at a concentration of 1:500 (goat anti-mouse $IgG_1$-488, A-21121), and then incubated with DAPI (1:100,000, Invitrogen, D1306) in PBS for 5 min. Slides were washed three times in PBS, then coverslipped with an in-house mounting media [20 mM Tris (pH 8.0), 90% glycerol and 0.5% N-propyl gallate]. Images were acquired using an Olympus FV3000 laser scanning confocal at 20× magnification, 10 μm z-stacks were acquired using a 0.5 μm step size. Images were max projected.

### Quantification and statistical analysis
All imaging and transport data were analyzed using GraphPad Prism version 10.42. RNA sequencing data were analyzed using edgeR (v4.6.3) and clusterProfiler (v4.17.0). Single-cell data were analyzed using Seurat (v5.3.0). An unpaired Student's t-test was used to analyze data sets with two groups. One-way ANOVAs were used to analyze data sets with more than two groups. Two-way ANOVAs were used to analyze data sets with two independent variables. Nested analyses were performed for sets of data using technical replicates. All data are mean±s.e.m.

### Acknowledgements
Text editing was performed using Grammarly. The authors subsequently reviewed and edited the content as necessary and take full responsibility for the publication's final content. We thank Duke's Division of Laboratory Animal Resources for expert animal care and husbandry, the Duke Light Microscopy Core Facility for providing access to imaging software, and Duke Computing Cluster for access to high-performance computing. We thank Juan Ramirez (Duke University, Durham, NC) for thoughtful discussions about bioinformatic approaches. We thank members of the Bagnat, Eroglu and Rawls labs for thoughtful discussion and comments on the manuscript. We also thank Medgenome for performing quality control and sequencing of bulk and scRNA sequencing libraries.

### Competing interests
The authors declare no competing or financial interests.

### Author contributions
Conceptualization: C.L.B., C.E., M.B.; Data curation: C.L.B., L.C., K.S., C.E., M.B.; Formal analysis: C.L.B., L.C., A.L.C., C.E., M.B.; Funding acquisition: C.E., M.B.; Investigation: C.L.B., L.C., T.R.L., C.E., M.B.; Methodology: C.L.B., L.C., K.S., T.R.L., D.S.L., L.C.F., C.M.J., V.Y.A., M.G., C.E., M.B.; Project administration: C.E., M.B.; Resources: L.C.F., C.M.J., V.Y.A., M.G., C.E., M.B.; Software: K.S.; Supervision: M.G., C.E., M.B.; Validation: C.L.B., M.G., C.E., M.B.; Visualization: C.L.B., L.C., D.S.L., C.E., M.B.; Writing – original draft: C.L.B., L.C., C.E., M.B.; Writing – review & editing: C.L.B., L.C., A.L.C., K.S., T.R.L., D.S.L., L.C.F., V.Y.A., M.G., C.E., M.B.

### Funding
This work was supported by the National Institute of Diabetes and Digestive and Kidney Diseases of the National Institutes of Health (R01DK137812 to M.B.), by a training grant awarded to Duke University from the National Institute of Diabetes and Digestive and Kidney Diseases (Duke University Training Grant in Digestive Diseases and Nutrition; 5T32DK007568 to C.L.B.) and by a National Eye Institute Center Core Grant for Vision Research P30 (EY005722 to V.Y.A.). C.E. is a Howard Hughes Medical Institute Investigator. Open Access funding provided by the Howard Hughes Medical Institute. Deposited in PMC for immediate release.

### Data and resource availability
All sequencing data generated in this study have been deposited in GEO. The developmental RNA sequencing data are available under accession number GSE301314, and are available for exploration at http://gutdevelopmentseq.org/; the Dab2 cWT versus cKO RNA sequencing data are available under accession number GSE301313; and the single cell sequencing data for mice are under accession number GSE30138. The raw data and code generated during this study can be accessed on Zenodo at https://doi.org/10.5281/zenodo.17251811. All other relevant data and details of resources can be found within the article and its supplementary information.

### Peer review history
The peer review history is available online at https://journals.biologists.com/dev/lookup/doi/10.1242/dev.205127.reviewer-comments.pdf

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
