## [Peer Review File · Development (Cambridge, England)]

The mouse neonatal small intestine is regionally specialized for protein absorption and transepithelial transport

Carina L. Block, Laura Childers, Abby L. Cortez, Kristina Sakers, Tylor R. Lewis, Daniel S. Levic, Lauren C. Frazer, Corey M. Jania, Vadim Y. Arshavsky, Misty Good, Cagla Eroglu and Michel Bagnat

DOI: 10.1242/dev.205127

Editor: James M Wells

Review timeline

Original submission:	23 July 2025
Editorial decision:	27 August 2025
First revision received:	3 October 2025
Accepted:	28 October 2025

Original submission

First decision letter

MS ID#: dev.205127

MS TITLE: The mouse neonatal small intestine is regionally specialized for protein absorption and transepithelial transport

AUTHORS: Carina Lea Block; Laura Childers; Abby Lynn Cortez; Kristina Sakers; Tylor R. Lewis; Daniel S. Levic; Lauren Carole Frazer; Corey M. Jania; Vadim Y. Arshavsky; Misty Good; Cagla Eroglu; Michel Bagnat

Dear Dr Bagnat,

I have now received all the referees' reports on the above manuscript, and have reached a decision. The referees' comments are appended below, or you can access them online: please go to: *****

As you will see, the referees express considerable interest in your work, but have some significant criticisms and recommend a substantial revision of your manuscript before we can consider publication. If you are able to revise the manuscript along the lines suggested, which may involve further experiments, I will be happy to receive a revised version of the manuscript. Your revised paper will be re-reviewed by one or more of the original referees, and acceptance of your manuscript will depend on your addressing satisfactorily the reviewers' major concerns. Please also note that Development will normally permit only one round of major revision. If it would be helpful, you are welcome to contact us to discuss your revision in greater detail. Please send us a point-by-point response indicating your plans for addressing the referees' comments, and we will look over this and provide further guidance.

Please attend to all of the reviewers' comments and ensure that you clearly highlight all changes made in the revised manuscript. Please avoid using 'Tracked changes' in Word files as these are lost in PDF conversion. I should be grateful if you would also provide a point-by-point response detailing how you have dealt with the points raised by the reviewers in the 'Response to Reviewers' box. If you do not agree with any of their criticisms or suggestions please explain clearly why this is so.

Reviewer 1

SUMMARY OF THE ADVANCE MADE IN THIS PAPER AND ITS POTENTIAL SIGNIFICANCE TO THE FIELD

The authors combine in vivo and ex vivo cargo transport assays with transcriptomic and genetic approaches in mice to uncover distinct roles for jejunal and ileal neonatal enterocytes. These findings show regional and functional specialization of enterocytes during early postnatal development and underscore conserved protein absorption mechanisms in vertebrates.

From this article, they provided evidence shown below.

They show that the jejunum is highly active in transepithelial transport, whereas the ileum specializes in their lysosomal degradation.

They clearly show that the endocytic system is distinct between the jejunum and ileum, in particular, during neonatal period.

The expression profiles of the jejunum and ileum is distinct as well which support the differences in their functions.

This expression profile is similar to the one in zebrafish, indicating the evolutionary conservation of functional differences in different parts of the intestine in vertebrates.

Dab2 is one of the major molecule involved in transcytosis of proteins, but not antibodies.

SUGGESTIONS TO AUTHORS

The authors present a number of data using different kinds of cutting-edge technologies to support their conclusions. Almost all of the experiments are well done and provide solid evidence.

However, this reviewer suggests several minor suggestions to improve their findings.

1) The authors show that the ileum is rather specialized for lysosomal degradation from morphological observations and transcriptional profiles. They seem to emphasize that this function of the ileum is supported by high expression of a complex of Dab2, Cubn and Amn as illustrated in Figure 5A. Therefore, it is expected that the function or/and morphology of the ileum will be affected most in the absence of these molecules. However, when they generate Dab2 conditional knockout, both endocytosis in the ileum and jejunum are similarly affected in Figure 5D and K. This reviewer needs some comments explaining this discrepancy.

2) Also, in Fig.5L, antibody uptake is upregulated in the cKO jejunum. The reviewer would appreciate it if the author could think of the reasons, if any.

3) Comparison of the intestine between mouse and zebrafish is interesting. However, many readers may not know where the LRE and IEC are localized. It is helpful to add cartoon like Fig.1A in the article by Childers et al (2025) to show where the LRE and IEC are.

4) Minor corrections: Fig.4G and Fig.4H should be corrected to Fig.3G and Fig.3H in p18. cWT should be corrected to WT in p21 and Fig.5.

Reviewer 2

SUMMARY OF THE ADVANCE MADE IN THIS PAPER AND ITS POTENTIAL SIGNIFICANCE TO THE FIELD

This manuscript from Block et al. describes regional differences in protein uptake/processing along the length to the neonatal mouse intestine and contrasts them with post-weaning intestine. This analysis includes uptake of exogenous tracers, quantification of degradation and transepithelial transport, and molecular analysis of differential gene expression.

Strengths of the manuscript include the differential analysis between the zebrafish and mouse enterocytes, demonstrating the conservation across species. In addition, the experiments utilizing the Dab2 knockout mouse are important in defining the different uptake and transepithelial transport regulators. The bulk and single cell RNA sequencing may yield insights into the regional specialization beyond what is currently known, but did not provide substantial new insights.

Overall, while the experiments appear to be carefully performed, the results are largely not novel.

SUGGESTIONS TO AUTHORS

1. There is a substantial literature that describes differences in uptake in the rodent neonatal jejunum and ileum. While some of it is cited in the text, work from Neutra and Gonnella and Abrahamson and Rodewald are not included. These studies show differential uptake in these regions and, in the case of some of the publications from Neutra, show transport of EGF and NGF across the neonatal ileum. Since one of the conclusions of this manuscript is that the ileum does not carry out transepithelial transport, these differences in findings need to be addressed.
2. In Figure 1K, the p value is given but the time point that is related to that p value is unclear. Presumably at 60 minutes?
3. With respect to the presence of LREs in neonatal human intestine, Moxey and Trier performed EM analysis on fetal human intestine and found that by 22 weeks gestational age the enterocytes appeared to be mature and lacked large lysosomes. The PAS staining in Figure 3 is not convincing and it would be useful to use a lysosomal marker. While I realize that these materials are limited, it would also be valuable to quantify the number of cells/villus that contain large vacuoles, as was done with the neonatal mice.
4. In addition, the timeline given for the age of the human neonates is confusing. 35 weeks postmenstrual age would be premature.

Reviewer 3

SUMMARY OF THE ADVANCE MADE IN THIS PAPER AND ITS POTENTIAL SIGNIFICANCE TO THE FIELD

This manuscript by Block et al. investigates the regional and functional specialization of jejunal and ileal enterocytes in the neonatal period. The manuscript finds through single cell and bulk sequencing, functional uptake assays, staining and genetically modified mice that jejunal and ileal enterocytes have different endolysosomal programs. The manuscript provides foundational information about unique roles of enterocytes in different regions of the small intestine during development, an important topic for the field. Overall, the manuscript is well written, and the data is straightforward and supports the major findings.

SUGGESTIONS TO AUTHORS

Major comments:

1. What is the expression pattern of Dab2, Cubn and Amn along the mouse intestine? Does Dab2 have a more prominent role in the ileum because it has higher baseline expression in the distal intestine compared to the mid intestine? The authors reported gene expression levels but are protein levels different in the small intestinal regions?
2. A methods section describing how data was analyzed is needed for the quantification graphs, for example Figure 1b has cells with vacuoles (% of total cells), is this per field of view? Are all cells counted for % of total cells ie non epithelial cells? Were multiple fields taken per mouse? What objective was used to acquire the images for the quantification? More information on how jejunum and ileum were defined for the experiments is needed as well. Different labs have widely varying definitions of the different intestinal segments.
3. More in depth discussion on neonatal endocytic mechanisms would bolster the paper. The authors mention redundancies in the endocytic machinery in the intestine during development but discussion of these varying modes and proteins involved such as Rab7, Syndapin 2, Syntaxins, munc18-2 would help in comparing and contrasting with the authors' major findings.
4. When Dab2 is conditionally deleted from the intestinal epithelium are other endocytic proteins up-regulated? If so which?

First revision

Author response to reviewers' comments

We would like to thank the reviewers for their overall positive assessment of our manuscript. In response to their insightful comments, we have included new analyses generated from published

human single-cell data sets, which are now included in figure 4. We have also made changes to the text and minor changes as requested. We feel these additions and changes have improved the manuscript. The changes are shown in the manuscript in blue font, and we respond to specific points made by the reviewers in blue font below.

Reviewer #1 (Comments to the Authors):

“The authors combine *in vivo* and *ex vivo* cargo transport assays with transcriptomic and genetic approaches in mice to uncover distinct roles for jejunal and ileal neonatal enterocytes. These findings show regional and functional specialization of enterocytes during early postnatal development and underscore conserved protein absorption mechanisms in vertebrates.

From this article, they provided evidence shown below. They show that the jejunum is highly active in transepithelial transport, whereas the ileum specializes in their lysosomal degradation.

- They clearly show that the endocytic system is distinct between the jejunum and ileum, in particular, during neonatal period.
- The expression profiles of the jejunum and ileum is distinct as well which support the differences in their functions.
- This expression profile is similar to the one in zebrafish, indicating the evolutionary conservation of functional differences in different parts of the intestine in vertebrates.
- Dab2 is one of the major molecule involved in transcytosis of proteins, but not antibodies.”

“The authors show that the ileum is rather specialized for lysosomal degradation from morphological observations and transcriptional profiles. They seem to emphasize that this function of the ileum is supported by high expression of a complex of Dab2, Cubn and Amn as illustrated in Figure 5A. Therefore, it is expected that the function or/and morphology of the ileum will be affected most in the absence of these molecules. However, when they generate Dab2 conditional knockout, both endocytosis in the ileum and jejunum are similarly affected in Figure 5D and K. This reviewer needs some comments explaining this discrepancy.”... “Also, in Fig.5L, antibody uptake is upregulated in the cKO jejunum. The reviewer would appreciate it if the author could think of the reasons, if any.”

We thank the reviewer for their kind and positive evaluation and for raising points that needed further clarification. Below, we provide our response and note that we have revised the discussion section of the manuscript to highlight these explanations.

Cargo uptake lies upstream of both lysosomal degradation and transcytosis. While previous studies have shown that both the jejunum and ileum can endocytose dietary proteins, it was not clear whether this uptake relies on the same non-selective receptor complex. To address this, we eliminated Dab2, a critical component of the Cubilin/Amnionless/Dab2 (Cubn/Amn/Dab2) scavenger receptor complex, in both regions of the intestine.

When we gavaged mCherry, a protein cargo that we had previously shown to be internalized via Cubilin/Amnionless/Dab2 in zebrafish, we observed impaired endocytosis in both the jejunum and ileum. This finding demonstrates that mCherry internalization depends on the same scavenger receptor complex in both regions.

We next tested horseradish peroxidase (HRP), another cargo without a dedicated receptor. Following Dab2 deletion, HRP transcytosis in the jejunum was reduced; however, this reduction reflects impaired protein uptake rather than a direct defect in transcytosis. By contrast, Dab2 deletion did not impair antibody transcytosis in the jejunum, which proceeds through the dedicated Fc receptor. Interestingly, we observed an increase in antibody transcytosis, which we interpret as a compensatory response to the loss of the non-selective uptake pathway.

Importantly, our quantitative assays allowed us to monitor the relative capacity in each intestinal region and thus better define their physiological functions.

“Comparison of the intestine between mouse and zebrafish is interesting. However, many readers may not know where the LRE and IEC are localized. It is helpful to add cartoon like Fig.1A in the article by Childers et al (2025) to show where the LRE and IEC are.”

We thank reviewer 1 for this suggestion, we have now added the recommended diagram, including a similar diagram for mice to Figure 4A.

“Fig.4G and Fig.4H should be corrected to Fig.3G and Fig.3H in p18.”

We thank the reviewer for catching this error we have updated the text with changes in blue.

“cWT should be corrected to WT in p21 and Fig.5.”

Genomic insertion of Cre recombinase can produce physiological effects on its own. To exclude unexpected off-target phenotypes, the recommended control in Cre-lox studies is to use Cre-only controls (Harno et al., 2013, *Cell Metab.*, PMID: 23823475). In our notation, we include a lowercase “c” before “WT” to indicate that control mice also carry a copy of Cre.

Reviewer #2 (Comments to the Authors):

“This manuscript from Block et al. describes regional differences in protein uptake/processing along the length to the neonatal mouse intestine and contrasts them with post-weaning intestine. This analysis includes uptake of exogenous tracers, quantification of degradation and transepithelial transport, and molecular analysis of differential gene expression. Strengths of the manuscript include the differential analysis between the zebrafish and mouse enterocytes, demonstrating the conservation across species. In addition, the experiments utilizing the Dab2 knockout mouse are important in defining the different uptake and transepithelial transport regulators. The bulk and single cell RNA sequencing may yield insights into the regional specialization beyond what is currently known, but did not provide substantial new insights.

Overall, while the experiments appear to be carefully performed, the results are largely not novel.”

We thank the reviewer for their thorough scrutiny of our manuscript and for raising several points that needed clarification. We are grateful for suggesting digging deeper into the human neonatal gut program. We were to identify published scRNAseq data that had not been analyzed with respect to the LRE program and were able to generate more compelling cross-species analyses. However, we respectfully disagree about the perceived novelty of findings and maintain that much of what was assumed to be established need in fact direct quantitative comparisons and, importantly, a molecular basis which we provide in our study.

“There is a substantial literature that describes differences in uptake in the rodent neonatal jejunum and ileum. While some of it is cited in the text, work from Neutra and Gonnella and Abrahamson and Rodewald are not included. These studies show differential uptake in these regions and, in the case of some of the publications from Neutra, show transport of EGF and NGF across the neonatal ileum. Since one of the conclusions of this manuscript is that the ileum does not carry out transepithelial transport, these differences in findings need to be addressed.”

We thank the reviewer for pointing out this previous work. We have now cited the recommended literature (pages 3, 4, 9, and 25). As noted by Reviewer #2, Gonnella et al. (1987) demonstrated that EGF is readily endocytosed by neonatal rat ileal enterocytes. Although most EGF is routed to lysosomal degradation, a small fraction is detected along the basolateral cell surface, indicating limited transepithelial transport. Importantly, EGF detected in serum was highly degraded, supporting that in the ileum the majority of EGF is processed through the lysosome.

Consistent with these findings, in Figure 1K of our manuscript, we observed an increase in HRP signal over time in the ileum. To test whether this increase was statistically significant, we performed a one-way ANOVA on HRP concentrations and found a significant rise beginning at 15 minutes. Thus, while the ileum carries out a small degree of transcytosis, its transport capacity remains substantially lower than that of the jejunum. Our findings align with previous reports of

limited transport but more clearly define regional differences in capacity.

“In Figure 1K, the p value is given but the time point that is related to that p value is unclear. Presumably at 60 minutes?”

In Figure 1K-L we performed a repeated measures two-way ANOVA with time and intestinal region as factors. The p-value displayed is for the main effect of region. This has now been updated in the legend.

“With respect to the presence of LREs in neonatal human intestine, Moxey and Trier performed EM analysis on fetal human intestine and found that by 22 weeks gestational age the enterocytes appeared to be mature and lacked large lysosomes. The PAS staining in Figure 3 is not convincing and it would be useful to use a lysosomal marker. While I realize that these materials are limited, it would also be valuable to quantify the number of cells/villus that contain large vacuoles, as was done with the neonatal mice.”

We thank the reviewer for this comment and the opportunity to address this concern. Respectfully, we note that Moxey and Trier (1979, PMID: 507402) present two images of a 22-week gestational intestine. The first image (Fig. 8A, included below, left) is from the proximal intestine and lacks the extensive apical canalicular system typically observed in neonatal enterocytes. The second image (Fig. 8B, included below, right) is from the distal intestine (presumably ileum) of the same fetus and shows both an apical canalicular system and a prominent lysosomal vacuole (L).

NOTE: Figure provided for reviewer has been removed. Reviewer Figure 2 showed Figure 8 from Moxey, P. C. and Trier, J.S. (1979) Development of villus absorptive cells in the human fetal small intestine: a morphological and morphometric study. *Anat Rec.* 195, 463-82. doi: 10.1002/ar.1091950307.

Reviewer Fig. 2. LREs are present in the distal intestine of a 22-week fetus. Figure legend from manuscript. **A)** VAC from proximal intestine of a 22-week fetus demonstrating marked increase in the smooth endoplasmic reticulum and the concomitant decrease in cytoplasmic glycogen, ATS and MCS. A single multivesicular body is present (X). **B)** The apical cytoplasm of a VAC from the distal intestine of the same fetus as in figure 8A. Little or no smooth endoplasmic reticulum is present and glycogen, MCS (L), and ATS (arrows) are still prominent.

MCS-meconium corpuscles; ATS-Apical tubular system; VAC- Villus absorptive cells

Figure 8 from Moxey and Trier, 1989

Given the limited availability of human tissue and the additional confounds associated with NEC patient samples, we could not perform an analysis parallel to our mouse data. Instead, we turned to two published single-cell RNAseq datasets that included ileal tissue from infants (Egozi et al., 2023, *PLoS Biol.*, PMID: 37205711) and pediatric patients (Elmentaite et al., 2020, *Dev Cell*, PMID: PMC7762816).

In the Egozi study, control infants underwent surgery for spontaneous ileal perforation (SIP). On average, these infants were born at 37 weeks of gestation and underwent surgery at 14 days of age. Fortunately, the dataset also included infants with necrotizing enterocolitis (NEC), who were born at ~28 weeks of gestation and underwent surgery at ~24.5 days of age.

As a comparison to our postweaning mice (P36), we incorporated control samples from Elmentaite et al. (2020). These pediatric patients underwent ileal biopsy but did not have Crohn's disease or other inflammatory intestinal conditions. Their ages ranged from 4 to 12 years, representing a broader developmental window than the infants.

In our analyses, we integrated infant and pediatric control samples and generated an object containing only enterocytes. These results are now included as Figure 4C in our revised manuscript. In addition to clustering by age, gene expression followed a developmental trajectory consistent with patterns predicted from our mouse RNA-sequencing data. Notably, *DAB2*, a

component of the endocytic complex, was expressed in 44% of infant enterocytes and at levels 2.5 log₂-fold higher in infants compared to pediatric samples. Furthermore, *PRDM1* and *MAMDC4*, two genes that drive the neonatal enterocyte program and are critical for formation of the neonatal enterocyte vacuole and apical canalicular system (Cox et al., 2018, *Cell Mol Gastroenterol Hepatol*, PMID: PMC5756061; Muncan et al., 2011, *Nat Commun.*, PMID: PMC3167062), were also expressed in a higher percentage of cells and at higher levels in infants. Conversely, genes associated with ileal maturation, which we predicted to be upregulated with age, were enriched in pediatric samples. By contrast, housekeeping genes remained stable across both groups. These data are now included as Fig. 4D.

To further investigate a conserved neonatal enterocyte program, we compared genes upregulated in infant enterocytes with those upregulated in the P6 mouse ileum. Pathway analysis revealed that infant enterocytes align with the neonatal mouse ileum in lysosomal and endosomal pathways. These results are now included as Figure 4E in our revised manuscript.

To determine whether a specific subset of infant enterocytes exhibits a stronger LRE signature, we re-clustered only the infant enterocytes. This analysis identified four clusters, with KEGG pathway analysis showing enrichment of lysosomal and endocytic pathways in cluster 0 (Fig. 4F- G). Consistent with this, *DAB2* expression was enriched in cluster 0 and largely overlapped with this population (Fig. S9A), suggesting that a subset of infant enterocytes retains the neonatal endocytic program.

Because most available infant intestinal tissue comes from patients recovering from NEC, who typically undergo extensive antibiotic treatment during recovery, which is known to alter intestinal maturation (Garcia et al., 2021, *Cell Mol Gastroenterol Hepatol.*, PMID: PMC8346670), we next asked whether NEC impacts the LRE program. Although we observed LREs in patient samples (Fig. 1), these were from infants after recovery from NEC, and LREs appeared less abundant than expected. This raised the possibility that NEC disrupts enterocyte specialization. To explore this possibility, we analyzed infant enterocytes from Egozi et al. (2023), comparing controls (spontaneous ileal perforation, SIP) with NEC. Control enterocytes clustered largely to the left of the UMAP in clusters 0 and 1, whereas NEC-derived enterocytes clustered to the right and represented the majority of cells in clusters 2 and 3 (Fig. S9 B-C). In NEC samples, the abundance of enterocytes in clusters 0 and 1 was reduced, with a shift toward clusters 2 and 3. These latter clusters showed enrichment for ribosomal and DNA replication pathways, consistent with accelerated epithelial turnover in NEC (Fig. 4G and Fig. S9C).

Together, these findings demonstrate that a subset of infant enterocytes retains a conserved LRE-like neonatal program, including the molecular machinery required for protein uptake. However, this program is largely lost in NEC, suggesting that NEC or its treatment remodels the intestinal epithelium and depletes functional LREs. This explains the low abundance of LREs in the NEC samples we analyzed. These new analyses are presented in Figures 4 and S8-S9 of our revised manuscript.

“The PAS staining in Figure 3 is not convincing and it would be useful to use a lysosomal marker.”

We appreciate this suggestion, and if additional tissue were available, we would perform staining in human infant samples. However, the tissue available to us was paraffin-embedded and required harsh antigen retrieval for immunofluorescence. Ultimately, this retrieval process compromised tissue integrity and made it difficult to distinguish true signal from autofluorescence.

“In addition, the timeline given for the age of the human neonates is confusing. 35 weeks postmenstrual age would be premature.”

Yes, this is correct. Premature birth is a known risk factor for NEC, and therefore most of the samples available to us are from preterm infants. The other sample presented in Figure 1 of the manuscript is from an infant at 58 weeks, and should therefore be more representative of a term infant intestine.

Reviewer #3:

“This manuscript by Block et al. investigates the regional and functional specialization of jejunal and ileal enterocytes in the neonatal period. The manuscript finds through single cell and bulk sequencing, functional uptake assays, staining and genetically modified mice that jejunal and ileal enterocytes have different endolysosomal programs. The manuscript provides foundational information about unique roles of enterocytes in different regions of the small intestine during development, an important topic for the field. Overall, the manuscript is well written, and the data is straightforward and supports the major findings.”

We thank the reviewer for their kind assessment and for their suggestions, which we have incorporated.

“What is the expression pattern of Dab2, Cubn and Amn along the mouse intestine? Does Dab2 have a more prominent role in the ileum because it has higher baseline expression in the distal intestine compared to the mid intestine? The authors reported gene expression levels but are protein levels different in the small intestinal regions?”

We have previously shown that Dab2 protein is highly abundant at the apical membrane of the neonatal mouse ileum (Reviewer Fig. 2A; Park et al., 2019, *Dev Cell*, PMID: PMC6783362). In addition, others have reported higher Dab2 expression in the neonatal rat ileum compared with the jejunum (Reviewer Fig. 2B; Vazquez-Carretero et al., 2010, *J Cell Biochem.*, PMID: 21080337). We have updated the revised manuscript to include a brief description of these findings with appropriate citations (page 13). Unfortunately, we have not identified reliable antibodies for Cubn or Amn, but based on gene expression data, we predict that Cubn protein would also be more highly expressed in the ileum than in the jejunum.

NOTE: Figure provided for reviewer has been removed. It showed Figure 7H from Park, J., Levic, D.S., Sumigray, K.D., Bagwell, J., Eroglu, O., Block, C.L., Eroglu, C., Barry, R., Lickwar, C.R., Rawls, J.F., Watts, S.A., Lechler, T. and Bagnat, M. (2019) Lysosome-Rich Enterocytes Mediate Protein Absorption in the Vertebrate Gut. *Dev Cell*. 51, 7-20.e6. doi: 10.1016/j.devcel.2019.08.001.

NOTE: Figure provided for reviewer has been removed. It showed Figure 5 from Vázquez-Carretero, M.D., García-Miranda, P., Calonge, M.L., Peral, M.J. and Ilundáin, A.A. (2011) Regulation of Dab2 expression in intestinal and renal epithelia by development. *J Cell Biochem*. 112, 354-61. doi: 10.1002/jcb.22931.

Reviewer Fig. 2. DAB2 protein levels are more abundant in the neonatal ileum. A) Immunofluorescence of DAB2 in the neonatal mouse intestine. Panel is from Figure 7 of Park et al., 2019. **B)** Western blots of Dab2 from isolated enterocyte apical membranes at P5 and P30 in rat jejunum, ileum and kidney cortex. Panel from Figure 5 of Vazquez-Carretero et al., 2011.

“A methods section describing how data was analyzed is needed for the quantification graphs, for example Figure 1b has cells with vacuoles (% of total cells), is this per field of view? Are all cells counted for % of total cells ie non epithelial cells? Were multiple fields taken per mouse? What objective was used to acquire the images for the quantification? More information on how jejunum and ileum were defined for the experiments is needed as well. Different labs have widely varying definitions of the different intestinal segments.”

We apologize for this oversight and have updated the methods to include a detailed description of how cells were counted and quantified (page 32). Furthermore, we have added a section detailing how intestinal segments were dissected (page 31)

“More in depth discussion on neonatal endocytic mechanisms would bolster the paper. The authors mention redundancies in the endocytic machinery in the intestine during development but discussion of these varying modes and proteins involved such as Rab7, Syndapin 2, Syntaxins, munc18-2 would help in comparing and contrasting with the authors' major findings.”

We thank the reviewer for this suggestion. We have assembled a more comprehensive panel of endocytic machinery in neonates, which is now included in the revised manuscript as Figure S4F

and referenced on page 13. In this analysis, we find that the neonatal jejunum shows selective enrichment for transport machinery mediating transcytosis of immunoglobulins and antigens (*Fcgrt*, *Pigr*, *Ly75*, *Anpep*, *Mfsd2a*), as well as regulators of apical-basolateral trafficking (*Rab8b*, *Rab30*, *Rilp*, *Yif1a*, *Ap1s3*, *Clint1*, *Arf4*, *Exoc3l4*). This profile supports intact cargo passage across the epithelium for systemic transport.

By contrast, the ileum is enriched for multi-ligand endocytic receptors and adaptors (*Lrp2*, *Cubn*, *Dab2*, *Emp2*, *Cav1*, *Podxl*, *Sdc1*), together with vesicle trafficking proteins (*Rab27b*, *Syt7*, *Syt15*, *Sort1*, *Reps2*, *Snx30*), consistent with the high-capacity endocytosis and lysosomal degradation characteristic of LREs. Many endocytic genes are expressed in both regions (e.g., *Rab31*, *Rab38*, *Rabep1*, *Snx8*, *Snx32*, *Wdr81*, *Clcn5*, *Dab2*, *Igf2r*, *Folr1/2*, *Scarb1*, *Icam1*, *Rdx*) and represent endosomal machinery and scavenger receptors that enable robust vesicle trafficking.

Together, these findings further emphasize the developmental division of labor we observe: the jejunum is specialized for selective transcytosis, whereas the ileum is specialized for bulk uptake and degradation.

“When *Dab2* is conditionally deleted from the intestinal epithelium are other endocytic proteins up-regulated? If so which?”

Yes, our original manuscript included panels Fig. 5H-I, which show an upregulation of clathrin- and receptor-mediated endocytosis in gene ontology analysis. Furthermore, the heatmap in Fig. 5I shows the upregulation of the alternative scavenger receptor megalin (*Lrp2*), as well as genes involved in clathrin-mediated endocytosis, including the clathrin light chain, *Clta*. These data point to a compensatory upregulation of canonical clathrin-mediated endocytosis in the *Dab2* cKO ileum.

Second decision letter

MS ID#: dev.205127R1

MS TITLE: The mouse neonatal small intestine is regionally specialized for protein absorption and transepithelial transport

AUTHORS: Carina Lea Block; Laura Childers; Abby Lynn Cortez; Kristina Sakers; Tylor R. Lewis; Daniel S. Levic; Lauren Carole Frazer; Corey M. Jania; Vadim Y. Arshavsky; Misty Good; Cagla Eroglu; Michel Bagnat

ARTICLE TYPE: Research Article

Dear Dr Bagnat,

I am happy to tell you that your manuscript has been accepted for publication in *Development*, pending our standard publication integrity checks.

Reviewer 1

SUMMARY OF THE ADVANCE MADE IN THIS PAPER AND ITS POTENTIAL SIGNIFICANCE TO THE FIELD

This reviewer is satisfied with the revisions made by the authors. This paper will advance our knowledge on the molecular mechanisms of endocytosis in different parts of the small intestine.

Reviewer 3

SUMMARY OF THE ADVANCE MADE IN THIS PAPER AND ITS POTENTIAL SIGNIFICANCE TO THE FIELD

The authors have sufficiently addressed my comments and in my opinion the comments of the other reviewers.